# Revisiting Learning-based Video Motion Magnification for Real-time Processing

**Hyunwoo Ha**[*]  
*Dept. of Electrical Engineering*  
*Pohang University of Science and Technology (POSTECH)*  
*hyunwooha@postech.ac.kr*

**Oh Hyun-Bin**[*]  
*Dept. of Electrical Engineering*  
*Pohang University of Science and Technology (POSTECH)*  
*hyunbinoh@postech.ac.kr*

**Kim Jun-Seong**  
*Dept. of Electrical Engineering*  
*Pohang University of Science and Technology (POSTECH)*  
*junseong.kim@postech.ac.kr*

**Kwon Byung-Ki**  
*Grad. School of Artificial Intelligence*  
*Pohang University of Science and Technology (POSTECH)*  
*byungki.kwon@postech.ac.kr*

**Kim Sung-Bin**  
*Dept. of Electrical Engineering*  
*Pohang University of Science and Technology (POSTECH)*  
*sungbin@postech.ac.kr*

**Linh-Tam Tran**  
*Kyung Hee University*  
*tamlt@khu.ac.kr*

**Ji-Yun Kim**  
*Dept. of Electrical Engineering*  
*Pohang University of Science and Technology (POSTECH)*  
*junekim@krafton.com*

**Sung-Ho Bae**  
*Kyung Hee University*  
*shbae@khu.ac.kr*

**Tae-Hyun Oh**  
*School of Computing*  
*Korea Advanced Institute of Science and Technology (KAIST)*  
*taehyun.oh@kaist.ac.kr*

**Reviewed on OpenReview:** *https://openreview.net/forum?id=TAmmPuExE1*

## Abstract

Video motion magnification is a technique to capture and amplify subtle motion in a video that is invisible to the naked eye. The deep learning-based prior work successfully models outstanding quality better than conventional signal processing-based ones. However, it still lags behind real-time performance, which prevents it from being extended to various online systems. In this paper, we revisit the first learning-based model and present experimental analyses, in particular on the identification of redundant components, the insertion of spatial bottlenecks, and the trade-off relationship between channel reduction and layer addition. By integrating the findings of each experiment, we present a real-time, deep learning-based motion magnification model that achieves a computational speed ranging from a minimum of **2.7×** to a maximum of **34.9× faster** than existing learning-based methods, while main-

---

[*]Both authors contributed equally to this research.

taining perceptually sufficient generation quality. To the best of our knowledge, this is the first learning-based motion magnification model that runs in real-time on Full-HD resolution videos even without ad hoc quantization. Project page: `https://fastdmm.github.io/`.

# 1 Introduction

Subtle motions, almost invisible to humans' naked eyes, are typically missed in videos. However, sensing such motions from the video is an irreplaceable modality for important applications, *e.g.*, visualizing the health status of infrastructures (Chen et al., 2014; 2015b;a; 2017) and buildings (Cha et al., 2017), sound (Davis et al., 2014), the motion of hot air (Xue et al., 2014), and medical signals like a human pulse (Balakrishnan et al., 2013; Tveit et al., 2016; Janatka et al., 2018). Video motion magnification (Liu et al., 2005; Wu et al., 2012; Wadhwa et al., 2013; 2014; Oh et al., 2018) is a technique that captures and amplifies such small motions to make them recognizable to the human eyes.

Video motion magnification is initially pioneered by signal processing based methods (Wu et al., 2012; Wadhwa et al., 2013; 2014; Zhang et al., 2017; Takeda et al., 2018; 2019; 2020; 2022). Unfortunately, these methods share fundamental limitations

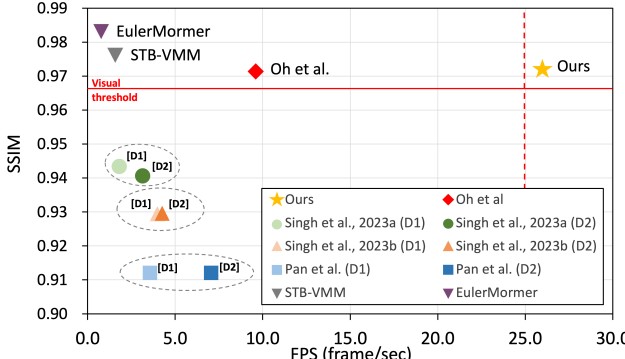

Figure 1: **Computational speed comparison of learning-based video motion magnification models.** Only our model exceeds the standard frame rate for real-time application (*i.e.*, 25 FPS), while achieving favorable motion magnification quality. The red line indicates the human-calibrated visual threshold, as defined in Sec. 4.2. See Table 3 for details.

derived from signal processing theories, as proved in Wu et al. (2012); Wadhwa et al. (2013), *e.g.*, noise sensitivity, magnification bounds, and limitations to occlusion/disocclusion of objects. Oh et al. (2018) tackle these limitations and demonstrate breakthrough results, by proposing the first learning-based model and the synthetic training dataset. Upon this success, subsequent works also take the learning-based approach (Singh et al., 2023a;b; Lado-Roigé & Pérez, 2023; Pan et al., 2024; Gao et al., 2022) and the training dataset from Oh et al. (2018) as a supervision signal (Singh et al., 2023a;b; Lado-Roigé & Pérez, 2023; Gao et al., 2022).

Despite the success of these learning-based methods, the prior methods do not meet real-time processing. As shown in Fig. 1, their computational speed is still below 10 frames per second (FPS) for Full-HD resolution videos even on decent GPUs. This limits their practical use in online applications, *e.g.*, structural vibration monitoring (An & Lee, 2022) and robotic surgery (Fan et al., 2021), where real-time processing is essential. In the existing real-time applications, the speed requirement is dealt with in an ad hoc way, *e.g.*, reducing input resolution, or by assuming that the core motion magnification algorithm would run in real-time (Fan et al., 2021).

In this regard, we aim to develop a real-time motion magnification model by revisiting Oh et al. (2018) as a baseline. We first analyze the components of the baseline model, examining their importance and functionality. The insights identified from these experimental analyses of the baseline model can be summarized as follows: 1) The encoder functions well even with a single linear layer, 2) Adding a spatial bottleneck on the decoder makes the model fast, and 3) The reduction of channels has a detrimental effect on the output quality, but this can be compensated by the addition of layers on the decoder.

Integrating these insights, we develop a real-time video motion magnification model, which achieves a computational speed ranging from at least **2.7×** to a maximum of **34.9× faster** than existing learning-based methods (Oh et al., 2018; Singh et al., 2023a;b; Lado-Roigé & Pérez, 2023; Pan et al., 2024), while maintaining generation quality comparable to the baseline (Oh et al., 2018) and showing favorable qualitative results relative to the recent transformer-based method (Lado-Roigé & Pérez, 2023).

To the best of our knowledge, this is the first learning-based motion magnification model that runs in real-time on Full-HD (FHD; 1920 × 1080) resolution videos with high-quality synthesis capability.

## 2 Related Work

Liu et al. (2005) pioneered Lagrangian video motion magnification, which relies on explicit motion estimation by optical flow and image warping. Later, Wu et al. (2012) coined the Eulerian approach that exploits intensity changes as a way of motion representation (Freeman et al., 1991a), becoming mainstream in motion magnification; thus, we focus on reviewing this line of work. The Eulerian approaches (Wu et al., 2012; Wadhwa et al., 2013; 2014; Zhang et al., 2017; Takeda et al., 2018; Oh et al., 2018; Takeda et al., 2019; 2020; 2022; Singh et al., 2023a;b; Lado-Roigé & Pérez, 2023; Pan et al., 2024) mainly consist of three stages: (a) Extracting a motion representation of each frame by a spatial decomposition; (b) Temporal filtering to capture motion of interest and amplifying the filtered signals with optional denoising; and (c) Recomposing the magnified frames from the amplified representations. Those works can be categorized according to main contributions: motion representation (Wu et al., 2012; Wadhwa et al., 2013; 2014; Oh et al., 2018; Takeda et al., 2020; Singh et al., 2023a;b; Lado-Roigé & Pérez, 2023; Pan et al., 2024) in (a) and (c), and temporal or denoising filters (Zhang et al., 2017; Takeda et al., 2018; 2019; 2022) in (b). Pillars of the motion magnification work established motion representations. Wu et al. (2012) model Eulerian motion representation by a first-order Taylor expansion, which results in applying Laplacian pyramid decomposition as a spatial decomposition. Subsequent works (Wadhwa et al., 2013; 2014) employ alternative wavelet filters, such as complex steerable pyramid (Freeman et al., 1991b). These hand-designed spatial decompositions are derived from traditional signal processing techniques, including the theories of polynomial, Fourier, and wavelet series, and show elegant modeling of motion as a shift of intensity signal. However, the modeling assumptions inherited by those signal processing theories limit their working regimes to pure translational motion and often yield artifacts in the regions where the theories do not support, *e.g.*, newly appearing or disappearing signals that frequently appear in real-world near occlusion/disocclusion of objects. Further, the magnitudes of their magnified motions are theoretically bounded as proved by themselves (Wu et al., 2012; Wadhwa et al., 2013).

To address these, Oh et al. (2018) propose the first learning-based approach that models motion representations by convolutional neural networks (CNN) and learns a spatial decomposition for motion magnification. They demonstrate significantly fewer artifacts, more robustness against noise and occlusion/disocclusion, and the linearity between the input amplification factor and the resulting magnified motion, which previous signal processing-based methods suffered from. Building on Oh et al. (2018), modifications in network architecture have been explored (Singh et al., 2023b; Lado-Roigé & Pérez, 2023; Pan et al., 2024; Wang et al., 2024a;b) to enhance the video frame generation performance of the learning-based method. However, all these endeavors have been underexplored from a computational perspective and are not feasible for real-time processing.

Distinguished from the above line of motion representations, another series of works propose temporal filters (Zhang et al., 2017; Takeda et al., 2018; 2022). They deal with degradation caused by large motion, noise, or drift, by designing dedicated temporal filters that change motion frequency of interest. Thus, their methods are effective for their targeted motions, but their behavior in other scenarios remains unclear. Also, they are mainly built on the motion representation of Wadhwa et al. (2013); thereby, they share the same limitations, a bit mitigated though. Therefore, their scope is independent of the motion representation works (Wu et al., 2012; Wadhwa et al., 2013; 2014; Oh et al., 2018; Takeda et al., 2020; Singh et al., 2023a;b; Lado-Roigé & Pérez, 2023; Pan et al., 2024; Wang et al., 2024a;b), including ours.

Recently, interest in real-time motion magnification has increased. Fan et al. (2021) present a robot surgery module that localizes blood vessels by magnifying pulsatile motion, but the method does not meet the real-time requirement despite their online application scenario. Singh et al. (2023a) propose a lightweight version of the baseline (Oh et al., 2018), but they fall far short of the standard frame rate for real-time applications (*i.e.*, 25 FPS) while exhibiting noticeable quality degradation.

In this regard, we aim to develop a real-time motion magnification model. To address this, we revisit and analyze Oh et al. (2018) as a baseline for two main reasons. First, as we can see in Fig. 1, Oh et al. (2018) demonstrate the fastest computation speed among existing learning-based methods while maintaining favorable quality, making it a logical starting point. Second, many recent learning-based models (Singh et al.,

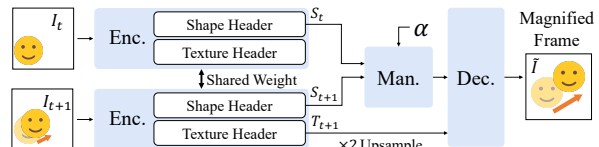

| Module | Input | Network Components | Output |
|---|---|---|---|
| (a) Enc. | $[H, W, 3]$ | Conv$\times$ 2, Resblk$\times$ 3, (d), (e) | See (d), (e) |
| (b) Man. | $[H/2, W/2, 32]$ | Conv$\times$ 2, Resblk$\times$ 1 | $[H/2, W/2, 32]$ |
| (c) Dec. | $[H/2, W/2, 64]$ | Conv$\times$ 2, Resblk$\times$ 9 | $[H, W, 3]$ |
| (d) Shape header | $[H/2, W/2, 32]$ | Conv$\times$ 1, Resblk$\times$ 2 | $[H/2, W/2, 32]$ |
| (e) Texture header | $[H/2, W/2, 32]$ | Conv$\times$ 1, Resblk$\times$ 2 | $[H/4, W/4, 32]$ |

Figure 2: **Overall architecture and specification of the baseline.** The baseline (Oh et al., 2018) consists of three modules: the encoder (Enc.), manipulator (Man.), and decoder (Dec.). Encoders are weight-shared. Given the two input frames, $I_t$ and $I_{t+1}$, the encoder takes each frame and outputs shape representation, $S_i$, and texture representation, $T_i$. The manipulator takes $S_t$ and $S_{t+1}$ and magnifies the motion by multiplying the amplification factor $\alpha$. The decoder reconstructs the magnified frame $\tilde{I}$.

2023a;b; Lado-Roigé & Pérez, 2023; Gao et al., 2022; Wang et al., 2024a;b) still adopt the training procedures and datasets introduced by Oh et al. (2018), establishing the work as the standard baseline.

## 3 Preliminary

In this section, we present preliminary backgrounds on video motion magnification and learning-based motion magnification and set up *the baseline.*

### 3.1 Video Motion Magnification

We first briefly describe the video motion magnification problem. To give intuition, we explain with an image intensity profile $f$ undergoing motion field $\delta(\mathbf{x}, t)$ over time $t$ as: $I(\mathbf{x}, t) = f(\mathbf{x} + \delta(\mathbf{x}, t))$, where $I(\mathbf{x}, t)$ is the observed intensity of an image (frame) at position $\mathbf{x}$ and time $t$. Then, the goal of motion magnification is to synthesize the motion magnified image $\tilde{I}(\mathbf{x}, t)$ as:

$$\tilde{I}(\mathbf{x}, t) = f(\mathbf{x} + (1 + \alpha)\delta(\mathbf{x}, t)), \tag{1}$$

where $\alpha$ is the amplification factor. In reality, the underlying image $f(\cdot)$ and motion $\delta(\cdot)$ are hidden and complicatedly entangled; thus, such manipulation requires decomposition of motion $\delta(\cdot)$ from $f(\cdot)$ which is fundamentally challenging.

To address this, as mentioned in Sec. 2, Eulerian motion magnification techniques (Wu et al., 2012; Wadhwa et al., 2013; 2014; Oh et al., 2018; Takeda et al., 2020; Singh et al., 2023a;b; Lado-Roigé & Pérez, 2023) apply spatial decompositions, *e.g.*, Laplacian pyramids (Wu et al., 2012) or CNN (Oh et al., 2018), to transform the image $I(\mathbf{x}, t)$ to a more favorable form to approximately separate motion information, *e.g.*, $I(\mathbf{x}, t) \rightarrow T(\mathbf{x}) + S(\delta(\mathbf{x}, t))$, where $T(\cdot)$ and $S(\cdot)$ represent texture and shape representations (Oh et al., 2018), respectively. Conceptually, the texture $T(\cdot)$ and the shape $S(\cdot)$ can be understood as proxy representations of the underlying profile $f(\cdot)$ and the residual depending on motion $\delta(\cdot)$, respectively.

The following procedure, extracting motion of interest on $S(\delta(\mathbf{x}, t))$ from the sum $T(\mathbf{x}) + S(\delta(\mathbf{x}, t))$, is typically implemented by temporal bandpass filters under the assumption that the motion signal of interest is within the passband of the filters. After extracting $S(\delta(\mathbf{x}, t))$, we can manipulate it to be $(1 + \alpha) \cdot S(\delta(\mathbf{x}, t))$ and add it back to $T(\mathbf{x})$, which allows to have the motion magnified image as

$$\tilde{I}(\mathbf{x}, t) \approx R[T(\mathbf{x}) + (1 + \alpha) \cdot S(\delta(\mathbf{x}, t))], \tag{2}$$

where $R[\cdot]$ denotes necessary reconstruction operations corresponding to respective spatial decompositions depending on the methods (Wu et al., 2012; Wadhwa et al., 2013; 2014; Oh et al., 2018; Takeda et al., 2020; Singh et al., 2023a;b; Lado-Roigé & Pérez, 2023) that revert back to image domain.

### 3.2 Baseline: Learning-based Motion Magnification

Our goal is to build an efficient model based on the deep motion magnification by Oh et al. (2018). The nature of modeling the motion magnification functions (*i.e.*, spatial decomposition, temporal filtering and magnification, and frame reconstruction) is reflected in the baseline as well, and those correspond to the encoder, manipulator, and decoder, respectively. For convenience, we refer to the input features of the decoder as *motion representations.*

We denote the baseline model as $\mathcal{G}(\cdot)$ and $I(\mathbf{x}, t) = I_t$ for simplicity. The model is learned to synthesize a magnified image $\tilde{I}(\mathbf{x}, t)$ from consecutive frames $\{I_t, I_{t+1}\}$ and an amplification factor $\alpha$ as input: *i.e.*, $\tilde{I}(\mathbf{x}, t) = \mathcal{G}(I_t, I_{t+1}, \alpha)$. Also, it can be generalized for multi-frames as $\mathcal{G}(\{I_t\}_{t=1}^N, \alpha)$ during inference time, where $N$ denotes a window size (refer to Fig. 2 for details).

One of the fundamental challenges in this field is that, unfortunately, there exist no ground-truth motion magnified videos in the real world to train the model. To address this issue, Oh et al. (2018) propose a synthetic training dataset, which is still adopted by most of the recent works (Singh et al., 2023a;b; Lado-Roigé & Pérez, 2023; Gao et al., 2022).

## 4 Experimental Setting

Before proceeding with the development, we analyze the baseline method to gain a better understanding of the learning-based motion magnification method. This section provides a detailed account of the experimental settings employed in the architecture design (Sec. 5) and results (Sec. 6).

### 4.1 Training and Evaluation Details

For training, we employ the synthetic dataset proposed by the baseline (Oh et al., 2018). This dataset includes pairs of consecutive frames, ground-truth magnified frames, and amplification factors. We split the dataset into training and validation sets, consisting of $95,000$ and $5,000$ data samples, respectively.

We utilize the validation set of the training dataset to evaluate the magnification quality of the models. In terms of metrics, we use image similarity measures, *e.g.*, the Structural Similarity Index Measure (SSIM). We also use Learned Perceptual Image Patch Similarity (LPIPS)[1] (Zhang et al., 2018) to evaluate perceptual aspects. We also measure architectural parameters (Params), floating-point operations per second (FLOPs), and frames per second (FPS) for inference wall-clock time on a single NVIDIA RTX 3090 GPU.

### 4.2 Calibration on Quality Evaluation Metric

Since similarity metrics (e.g., SSIM) measured on synthetic data often exhibit a domain gap with human perception in real-world scenarios, we conducted a human study to establish a reliable quality standard. We asked 20 participants to rate the visual similarity of magnified real videos compared to reference frames on a 0–5 scale. By correlating these scores with SSIM values across various model variants, we identified that a score of "3" (Adequate Similarity) corresponds to an SSIM of approximately 0.966. Based on this, we establish a *visual threshold* of 0.966 as the primary criterion for acceptable generation quality. Detailed experimental setups, participant instructions, and calibration results are provided in Appendix H.

### 4.3 Tests for Noise Handling

Effective noise handling is critical for separating subtle motion from artifacts (Wu et al., 2012). Following prior protocols (Oh et al., 2018), we conduct two complementary tests on synthetic data: a *noise test* and a *subpixel test*, where the latter evaluates motion magnitudes from 0.01 to 1 pixel. The noise test evaluates robustness against varying noise levels. However, since a model might achieve high SSIM by merely denoising (replicating inputs) rather than magnifying motion, the subpixel test is essential. It verifies the capability to isolate and magnify minute motions to a target of 10 pixels, ensuring the model accurately amplifies motion instead of simply suppressing small changes.

## 5 Architecture Design

We analyze the baseline model (Oh et al., 2018) to identify redundant components and architectural bottlenecks, which helps clarify the crucial characteristics for the video motion magnification task. Our analyses are threefold: 1) learning-to-remove redundant components (Sec. 5.1), 2) spatial bottleneck for frame reconstruction (Sec. 5.2), and 3) channel reduction with adding layers (Sec. 5.3). By integrating these insights, we introduce a real-time learning-based motion magnification architecture (Sec. 5.4).

---

[1]We measure LPIPS using the VGG16 network (Simonyan & Zisserman, 2014).

### 5.1 Learning-to-Remove Redundant Components

Oh et al. (2018) reported an interesting finding that linear approximations of their shape encoder resemble manually-designed linear filters in signal processing (Wu et al., 2012; Wadhwa et al., 2013), *e.g.*, directional edge detector, Laplacian operator, and corner detector. From the observation of resemblance with linear filters, we pose a question, "Do we really need non-linear layers in the encoder?". This motivates us to conduct a component ablation experiment to analyze the importance or redundancy of neural components. By identifying these components, we can find better trade-offs between computational efficiency and quality, facilitating the deployment in resource-constrained environments without compromising prediction accuracy.

Inspired by Dror et al. (2021) and Huang & Wang (2018), we design a *learning-to-remove method* that uses a learnable switch parameter to drive a layer toward a simpler state without unnecessary components. The method begins with parameterizing around the component of interest as:

$$F(x) = (1 - \omega)A_o(x) + \omega A_t(x), \quad 0 \leq \omega \leq 1, \tag{3}$$

where $\omega$ is a learnable switch parameter, $A_o(\cdot)$ and $A_t(\cdot)$ are element functions, *e.g.*, layers. $A_o(\cdot)$ is an original function standing for a pre-existing state, and $A_t(\cdot)$ a target function standing for a desirable state after the removal, *i.e.*, a much simpler element function. Initially, the learnable parameter $\omega$ is set to zero, so that $F(x) = A_o(x)$. If the learnable parameter $\omega$ approaches 1, then $F(x) \to A_t(x)$. Along with this parameterization, we aim to remove and analyze unnecessary components while preserving the task quality. We fine-tune the target architecture, *i.e.*, the baseline, with the learnable parameter $\omega$ by minimizing the following objective function,

$$\mathcal{L}_{total} = \mathcal{L}_{task} + \lambda_{rm}\mathcal{L}_{rm}, \tag{4}$$

where $\mathcal{L}_{task}$ denotes the original motion magnification task loss, $\mathcal{L}_{rm}$ the bias term, and $\lambda_{rm}$ a loss weight. We define the term $\mathcal{L}_{rm}$ as:

$$\mathcal{L}_{rm} = \frac{1}{K}\sum\nolimits_{k \in K}(1 - \omega_k^p), \tag{5}$$

where $p$ is a hyperparameter regulating the gradient of near $\omega = 1$, and $K$ the number of components. Equation (5) encourages the parameter $\omega$ close to 1 during optimization; thereby minimizing $\mathcal{L}_{total}$ finds the parameters $\{\omega\}$ that balance the task quality and component removal. By fine-tuning the whole neural network with $\mathcal{L}_{total}$, we distinguish which components are redundant to perform the task. After convergence, we explicitly discard the components whose resulting $\omega$ is larger than a threshold $\tau$. Subsequently, we train the processed model using only $\mathcal{L}_{task}$ to achieve the highest quality under the filtered configuration.

Using the learning-to-remove method, we conduct an experiment that can reveal which module is relatively unimportant and can be substituted with a linear one. Nine configurations were established by applying the learning-to-remove method to only one module (encoder, manipulator, or decoder) and one type of component (activation functions, skip connections, or residual blocks) for each configuration. In this experiment, we only investigate the components in residual blocks since they account for over 84% of the parameters in the baseline model. The experimental results are summarized in Table 1 and we discuss them in the following paragraphs. In addition, we also show that the learning-to-remove method can be successfully adapted to recent video motion magnification architectures (Lado-Roigé & Pérez, 2023; Wang et al., 2024a) and another task with inhomogeneous architectures such as image super resolution (Lim et al., 2017). Please see Appendix C for the detailed results.

**Encoder** Removing all ReLUs and skip-connections of the residual blocks in the encoder results in *a negligible drop in quality*. We further find that the complete removal of the non-linearity from the encoder does not show a significant difference in quality. We visualize the estimated impulse response of the shape encoder of the baseline and ours in Fig. 3, showing that our linear encoder behaves similarly to the deep non-linear baseline (Oh et al., 2018). Moreover, removing all the residual blocks significantly reduces the number of parameters and FLOPs, while resulting in negligible drops in quality.

**Manipulator** The result for the manipulator is analogous to the encoder; *i.e.*, all the components in the residual block can be removed with almost no quality loss. As a separate test, we manually remove the ReLU behind the convolutional layer of the manipulator, which is outside of the residual block. This results

Table 1: **Experiments with the component removal in residual blocks for each module.** We experiment with removal of each component (*i.e.*, ReLU, skip-connection of a residual block, and a whole residual block) module-by-module (*i.e.*, the encoder, the manipulator, and the decoder). All the experiments are conducted using a pre-trained baseline model (300 epochs). The cross mark ✗ indicates that the components are completely removed, and the triangle symbol △ denotes that a few components survive.

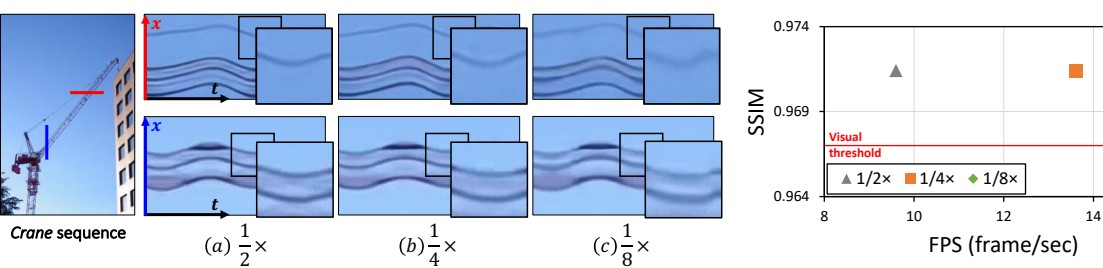

Figure 3: **Estimated impulse response of shape encoder.** We visualize the estimated kernels of the shape encoder in our (linear) model and the baseline (Oh et al., 2018) (non-linear) model. The kernels of the baseline encoder (top) and our deep linear neural encoder (bottom) have similar functionality to the conventional linear filters.

| Module | ReLU | Skip | Block | # Params [K] ↓ | FLOPs [G] ↓ | Quality Metric | |
|---|---|---|---|---|---|---|---|
| | | | | | | SSIM ↑ | LPIPS ↓ |
| Baseline | – | – | – | 967 | 41.3 | 0.980 | 0.180 |
| Encoder | ✗ | – | – | 967 | 41.3 | 0.979 (-0.001) | 0.187 (+0.007) |
| | – | ✗ | – | 967 | 41.3 | 0.977 (-0.003) | 0.195 (+0.015) |
| | – | – | ✗ | 838 (-129) | 37.6 (-3.7) | 0.978 (-0.002) | 0.186 (+0.006) |
| Manipulator | ✗ | – | – | 967 | 41.3 | 0.979 (-0.001) | 0.181 (+0.001) |
| | – | ✗ | – | 967 | 41.3 | 0.980 (-0.000) | 0.181 (+0.001) |
| | – | – | ✗ | 948 (-19) | 40.6 (-0.7) | 0.979 (-0.001) | 0.182 (+0.002) |
| Decoder | △ | – | – | 967 | 41.3 | 0.966 (-0.014) | 0.252 (+0.072) |
| | – | △ | – | 967 | 41.3 | 0.971 (-0.009) | 0.231 (+0.051) |
| | – | – | △ | 524 (-443) | 25.0 (-16.3) | 0.945 (-0.035) | 0.317 (+0.137) |

Figure 4: [**Left**] Qualitative results varying the spatial resolution of the representations, [**Right**] **Quantitative results varying the spatial resolution of the representations.** We choose two lines in the original frame and plot an **x**-t slice for the *crane* sequence. $\frac{1}{2}\times$ denotes the baseline model (Oh et al., 2018). We notice that the differences of qualitative results between $\frac{1}{2}\times$ and $\frac{1}{4}\times$ are almost unnoticeable. In contrast, $\frac{1}{8}\times$ shows more blurry results than the others. We also measure SSIM and FPS results varying the different downsampling factors and plot the SSIM versus FPS trade-off graph. The $\frac{1}{4}\times$ downsampling significantly gains FPS with marginally sacrificing SSIM.

in degraded quality without any computational benefit. This is consistent with the report by the baseline that the manipulator without ReLU blurs strong edges and is more prone to noise.

**Decoder** Unlike the encoder and manipulator, removing any component from the decoder noticeably reduces quality, with SSIM dropping below the visual threshold when elements like ReLU or residual blocks are removed; even if some components remain intact. This highlights the critical role, *frame reconstruction*, of the decoder's components in maintaining output quality.

## 5.2 Spatial Bottleneck for Frame Reconstruction

The decoder is crucial for reconstructing high-resolution outputs from compressed features. As established in Sec. 5.1, the decoder is significantly more sensitive to component removal than the encoder or manipulator. We attribute this to the decoder's requirement for a large receptive field to effectively integrate spatial information for intricate magnified outputs. Reducing model depth (*i.e.*, removing layers or blocks) shrinks this receptive field, thereby limiting the decoder's capability to integrate information over larger spatial extents, which is essential for high-quality motion magnification.

| Structural change | FLOPs [G] ↓ | FPS ↑ | SSIM ↑ | LPIPS ↓ |
|---|---|---|---|---|
| Baseline (Oh et al., 2018) | 41.3 | 9.6 | 0.971 | 0.190 |
| + 1/4× Spatial Resolution | 24.6 | 13.6 | 0.971 | 0.203 |
| + Single Linear Encoder | 20.5 | 16.8 | 0.970 | 0.208 |
| + Scaling $d_D$ and $c_D$ | 20.5 | 16.7 | 0.973 | 0.196 |
| + Scaling $C_{\text{upsample}}$ | 12.8 | 21.4 | 0.971 | 0.206 |
| + Scaling $c_E$ and $c_{\text{downsample}}$ | 9.86 | 25.8 | 0.971 | 0.211 |
| **Training loss change** | **FLOPs [G] ↓** | **FPS ↑** | **SSIM ↑** | **LPIPS ↓** |
| + Perceptual Loss (proposed) | **9.86** | **25.8** | **0.972** | **0.163** |

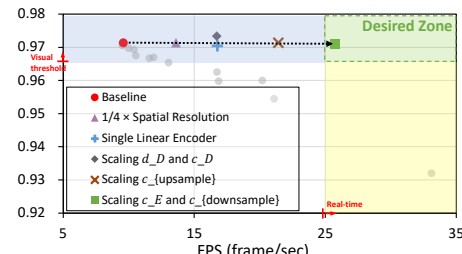

Figure 5: **Ablation study on structural change and training loss change in model architecture.** We progressively impose structural changes on the baseline model (Oh et al., 2018), reaching our proposed lightweight model. Frames Per Second (FPS) is calculated for input frames of resolution $1920 \times 1080$, while the other metrics are measured on synthetic data with $384 \times 384$. We denote by $d_D$ the decoder depth, by $c_D, c_E$ the channel dimensions, and by $c_{\text{upsample}}, c_{\text{downsample}}$ the up/downsampling layer dimensions.

The baseline (Oh et al., 2018) downsamples the spatial resolution of motion representation by $\frac{1}{2}\times$ at the beginning of the encoder, and upsamples it at the end of the decoder. This spatial bottleneck is designed to decrease memory footprint and increase the receptive field size, which is crucial for effective frame reconstruction in motion magnification. However, this reduction in feature size can result in information loss, adversely affecting the quality of the motion-magnified frame.

To address this trade-off between computational efficiency and magnification quality, we develop two alternative architectures with downsampling rates of $\frac{1}{4}\times$ and $\frac{1}{8}\times$, respectively. We then measure the trade-off between SSIM and FPS. As shown in Fig. 4, the $\frac{1}{4}\times$ setting shows qualitative results and SSIM comparable to those of the baseline $\frac{1}{2}\times$ setting, while performing $1.85\times$ faster FPS than the baseline. The $\frac{1}{8}\times$ downsampled representation further accelerates the computational speed, but suffers significant quality degradation, producing artifacts and much blurrier magnified frames compared to the $\frac{1}{4}\times$ one. Given these findings, the $\frac{1}{4}\times$ downsampled representation appears to be the better choice for achieving an efficient model. It offers a substantial gain of computational speed with only an acceptable loss in quality, making it a viable solution for balancing computational efficiency with magnification quality. We provide further detailed analyses of small motion and noise handling capabilities of downsampling motion representations in Appendix G.

### 5.3 Channel Reduction with Adding Layers

While reducing channel dimensions is a common strategy for lightweight networks (Singh et al., 2023a;b), naïvely applying it to the baseline (Oh et al., 2018) leads to significant degradation in generation quality. To address this, we propose increasing the network depth ($d_D$) while reducing channel dimensions ($c_D$) to maintain constant FLOPs. This approach aims to compensate for the quality loss by increasing the receptive field size. We conduct a controlled experiment varying $d_D$ and $c_D$ inversely.

Table 2: Controlled experiment on the impact of varying layer depth $d_D$ in the decoder. Underlined: baseline configuration. Bold: best SSIM, LPIPS.

| $d_D$ | $c_D$ | FLOPs (G) | SSIM ↑ | LPIPS ↓ |
|---|---|---|---|---|
| 3 | 112 | 42.3 | 0.962 ($-0.009$) | 0.232 ($+0.041$) |
| 6 | 80 | 42.7 | 0.969 ($-0.002$) | 0.202 ($+0.011$) |
| 9 | 64 | 41.3 | 0.971 | 0.191 |
| 12 | 56 | 42.1 | 0.972 ($+0.001$) | 0.186 ($-0.005$) |
| 15 | 48 | 40.0 | 0.973 ($+0.002$) | 0.183 ($-0.008$) |
| **36** | **32** | 41.5 | **0.975** ($+0.004$) | **0.172** ($-0.019$) |
| 144 | 16 | 41.6 | 0.974 ($+0.003$) | 0.175 ($-0.016$) |

Table 2 indicates that networks with deeper depth generally achieve higher SSIM and lower LPIPS values. We empirically observe that extreme reduction ($[d_D, c_D] = [144, 16]$) performs less effectively than $[36, 32]$, suggesting that a minimum channel dimension is required. Based on this, we adopt the $[36, 32]$ configuration and proportionally reduce the encoder channels ($32 \rightarrow 16$) to maximize computational efficiency without significant quality loss.

### 5.4 Proposed Architecture

As illustrated in Fig. 5, we ablate the architecture including three key changes: (i) we downsample the spatial resolution of the latent motion representation in the decoder; (ii) we simplify the encoder to a single

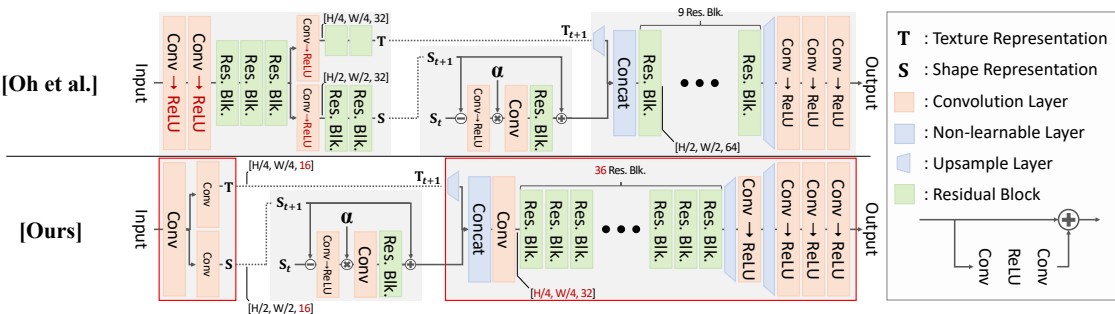

Figure 6: **Overall architecture of the proposed networks.** We present a real-time deep learning-based motion magnification model. The dimension of representations are denoted as [height, width, channel].

linear layer[2]; and (iii) we scale the decoder's residual blocks in layer depth ($d_D$) and channel dimension ($c_D$), keeping $c_E$ fixed. Consistent with the controlled experiment in Table 2, varying $d_D$ and $c_D$ alone has little impact on FLOPs. However, reducing the channel dimension of the upsampling convolutions, $c_{\text{upsample}}$, to align it with the decoder width $c_D$, yields additional FLOPs savings. Similarly, we can also reduce $c_E$ and downsampling convolutions $c_{\text{downsample}}$ without significant quality degradation.

In addition, we add perceptual loss using VGG16 (Zhang et al., 2018) (not included in our architectural ablation) to the training loss to improve perceptual quality[3] [4].

By integrating these findings, we develop a deep learning-based motion magnification model as shown in Fig. 6 that achieves real-time speed while maintaining favorable magnification quality.

## 6 Results

In this section, we assess the quality and effectiveness of our proposed model by comparing it with previous methods. Initially, we evaluate our model's quantitative performance on the synthetic dataset proposed by the baseline (Oh et al., 2018) (Sec. 6.1). In addition, we examine the methods' ability to handle noise and magnify subpixel motion (Sec. 6.3). Next, we present the qualitative results of our model and other motion magnification methods across various aspects: (1) Qualitative results on real video samples (Sec. 6.2), (2) Compatibility with temporal filter (Sec. 6.2), and (3) Physical accuracy of motion magnification (Sec. 6.4).

### 6.1 Quantitative Analysis

We conduct a quantitative evaluation of our model and other learning-based methods regarding computational efficiency and the quality of generated frames. Table 3 shows that only our proposed model surpasses the standard frame rate of real-time applications (*i.e.*, 25 FPS), while achieving favorable motion magnification quality. Our model achieves computational speed ranging from at least 2.7× to a maximum of 34.9× faster than other learning-based methods (Oh et al., 2018; Singh et al., 2023a;b; Lado-Roigé & Pérez, 2023; Pan et al., 2024; Wang et al., 2024a;b). Overall, our method achieves the second-best LPIPS score among the methods, while having a lower parameter count compared to the baseline (Oh et al., 2018) and the recent methods (Lado-Roigé & Pérez, 2023; Pan et al., 2024; Wang et al., 2024a;b). We also report a quantitative comparison on more realistic data in Appendix A.

In contrast, the other methods either lag behind ours in both generation quality and computational speed (Singh et al., 2023a;b; Pan et al., 2024), or exhibit comparable generation quality but are significantly slower (Oh et al., 2018; Lado-Roigé & Pérez, 2023; Wang et al., 2024a). Specifically, Singh et al. (2023a) achieve the lowest number of parameters compared to other methods, but this does not translate

---

[2]After removing all nonlinear activations and residual blocks in the encoder, the remaining CNN layers naturally collapse into one layer, slightly reducing FLOPs without compromising quality.

[3]While Oh et al. (2018) reported that adding the perceptual loss does not improve SSIM, we found that using the perceptual loss is effective qualitatively as we shall see in the result.

[4]Including the perceptual loss increases training time from 7.2 to 25.6 hours on a single RTX 3090 GPU.

Table 3: **Quantitative results.** We provide the detailed quantitative comparison aligned with Fig. 1. Frames Per Second (FPS) is calculated for input frames of resolution $1920 \times 1080$, while the other metrics (*i.e.*, SSIM and LPIPS) are for resolution $384 \times 384$. For studies that provide both a main model and a lightweight variant, we denote the main model as *D1* and the lightweight model as *D2*. **Bold** indicates the best performance, and underlined indicates the second-best performance.

| Model | FPS ↑ | # Params [M] ↓ | SSIM ↑ | LPIPS ↓ |
|---|---|---|---|---|
| Oh et al. (2018) | 9.6 | 0.967 | 0.971 | 0.190 |
| Pan et al. (2024) (D2) | 7.0 | 17.288 | 0.912 | 0.298 |
| Singh et al. (2023b) (D2) | 4.2 | **0.054** | 0.930 | 0.308 |
| Singh et al. (2023b) (D1) | 4.0 | 0.117 | 0.930 | 0.279 |
| Pan et al. (2024) (D1) | 3.5 | 17.288 | 0.912 | 0.298 |
| Singh et al. (2023a) (D2) | 3.1 | 0.185 | 0.941 | 0.308 |
| FD4MM (Wang et al., 2024b) | 2.2 | 1.47 | – | – |
| Singh et al. (2023a) (D1) | 1.8 | 1.814 | 0.943 | 0.265 |
| STB-VMM (Lado-Roigé & Pérez, 2023) | 1.6 | 31.335 | 0.977 | 0.176 |
| EulerMormer (Wang et al., 2024a) | 0.7 | 1.51 | **0.984** | **0.107** |
| Ours | **25.8** | 0.731 | 0.972 | 0.163 |

into increased computational speed. Computational costs on diverse hardware, detailed runtime profiling, and additional speed–resolution trade-off analyses are provided in Appendix I.

## 6.2 Motion Magnification on Real Video Samples

We test our method on real-world videos, including mechanical vibrating units (AC2) and structural components (Column, Wheel), to demonstrate practical applicability.

Figure 7 includes two general processing modes in learning-based motion magnification: *Static*, which uses a fixed reference $(X_1, X_t)$, and *Dynamic*, which employs a sliding reference $(X_{t-1}, X_t)$. In both cases, the model magnifies the difference between the reference frame and $X_t$ frame without advanced temporal filters. Figure 7 also includes the case when applying temporal filters (*e.g.*, Finite Impulse Response (FIR), butterworth). In addition to the qualitative evidence from the x-t slices and the supplementary video, a quantitative long-term temporal stability analysis is provided in Appendix B.

**Advantages of learning-based approaches.**   When comparing the phase-based method (Wadhwa et al., 2013) with other learning-based methods (Oh et al., 2018; Pan et al., 2024; Singh et al., 2023a;b; Lado-Roigé & Pérez, 2023; Wang et al., 2024a;b), a notable disparity in generation quality becomes apparent. Learning-based methods produce relatively higher-quality results and demonstrate superior handling of edge cases, such as occlusion or disocclusion, compared to the phase-based method, which exhibits noticeable ringing artifacts and produces blurry results.

**Comparison with the baseline.**   Our model shows generation quality comparable to that of the baseline model (Oh et al., 2018), while running in real-time on Full-HD videos. Our model also captures the edges clearly and shows no ringing artifacts as well as the baseline.

**Comparison with other learning-based methods.**   Overall, many learning-based methods (Pan et al., 2024; Singh et al., 2023a;b), except for recent transformer-based architectures (Lado-Roigé & Pérez, 2023; Wang et al., 2024a;b), tend to produce results with more noise, blur, and ringing artifacts compared to our model and the baseline. Notably, STB-VMM occasionally exhibits blocking artifacts, when processing lower-resolution video sequences such as the *baby* sequence with a resolution $960 \times 544$. The artifacts may arise from its use of a patch-wise processing technique, a characteristic of Vision Transformer architecture (Dosovitskiy et al., 2020; Liu et al., 2021). In contrast, our method is less susceptible to these artifacts and degradation, while uniquely achieving real-time processing speed in Full-HD among all the learning-based methods.

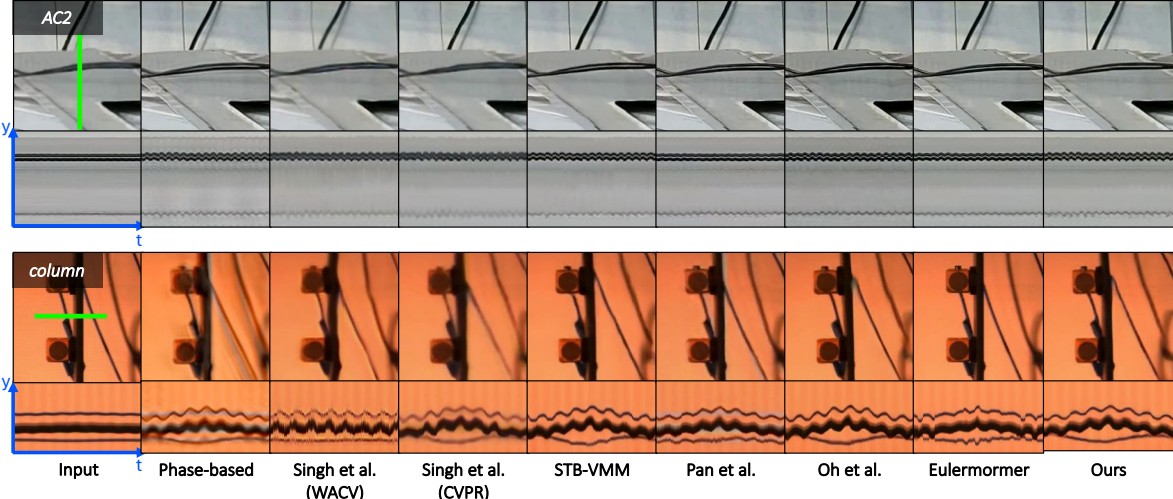

(a) Qualitative comparison across models (Top: *static*, Bottom: *dynamic*)

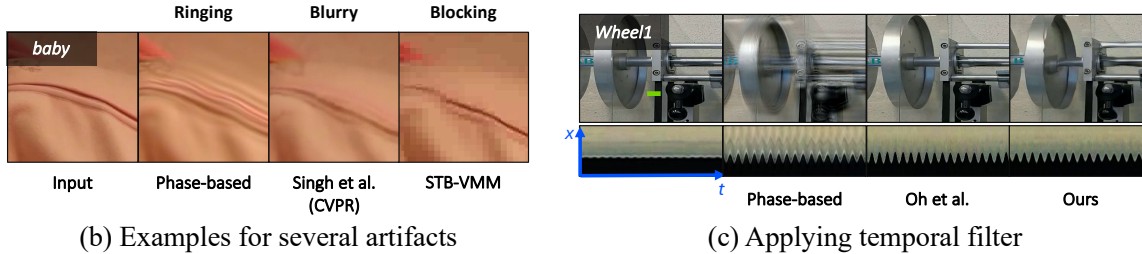

(b) Examples for several artifacts      (c) Applying temporal filter

Figure 7: **Motion magnification results.** (a) Qualitative comparison across models (Top: *Static* mode, Bottom: *Dynamic* mode), (b) Examples for several artifacts, and (c) Results applying temporal filter. Ours successfully magnifies the given video sequences without causing any artifacts even without temporal filters, and also shows compatibility with temporal filters. Configurations can be found in Appendix E.

**Compatibility with Temporal Filter.** Although our model is capable of producing fine results without relying on temporal filtering, incorporating such filtering can further reduce artifacts caused by unwanted motions, as suggested in Oh et al. (2018). By directly applying temporal filtering to motion representations, we effectively isolate only the desired motion within a temporal context, before feeding it into the decoder. Our model reconstructs a clear periodic signal, demonstrating its compatibility with temporal filters, similar to the baseline (Oh et al., 2018). The phase-based method (Wadhwa et al., 2013) can also capture periodic signal, however, it tends to produce severe ringing artifacts.

## 6.3 Noise and Subpixel Test

We perform a quantitative evaluation comparing our model and other motion magnification methods (Wadhwa et al., 2013; Oh et al., 2018; Singh et al., 2023a;b; Lado-Roigé & Pérez, 2023; Pan et al., 2024) in handling subpixel motion and noise, following the evaluation setup described in Sec. 4.3. As shown in Fig. 8, our model demonstrates noise handling capability comparable or superior to STB-VMM (Lado-Roigé & Pérez, 2023) and the baseline model in both small motion and large motion noise tests. Notably, in the small motion case, ours surpasses STB-VMM (Lado-Roigé & Pérez, 2023) as the noise factor becomes greater than 1.

Singh *et al.* (Singh et al., 2023a;b) exhibit distinctive noise characteristics that differ from other motion magnification methods. Specifically, they show weaker noise handling capability under a noise factor of 0.1 in the small motion noise test. While it may appear that their methods outperform others as the noise factor increases, this perception is deceptive. As shown in Fig. 8 [Bottom], Singh *et al.* (Singh et al., 2023a;b) produce output frames that closely resemble the input frames rather than the ground truth (GT) frames. This suggests these methods often merely replicate the input image instead of effectively magnifying small motions. On the other hand, our model, along with Oh et al. (2018) and STB-VMM (Lado-Roigé &

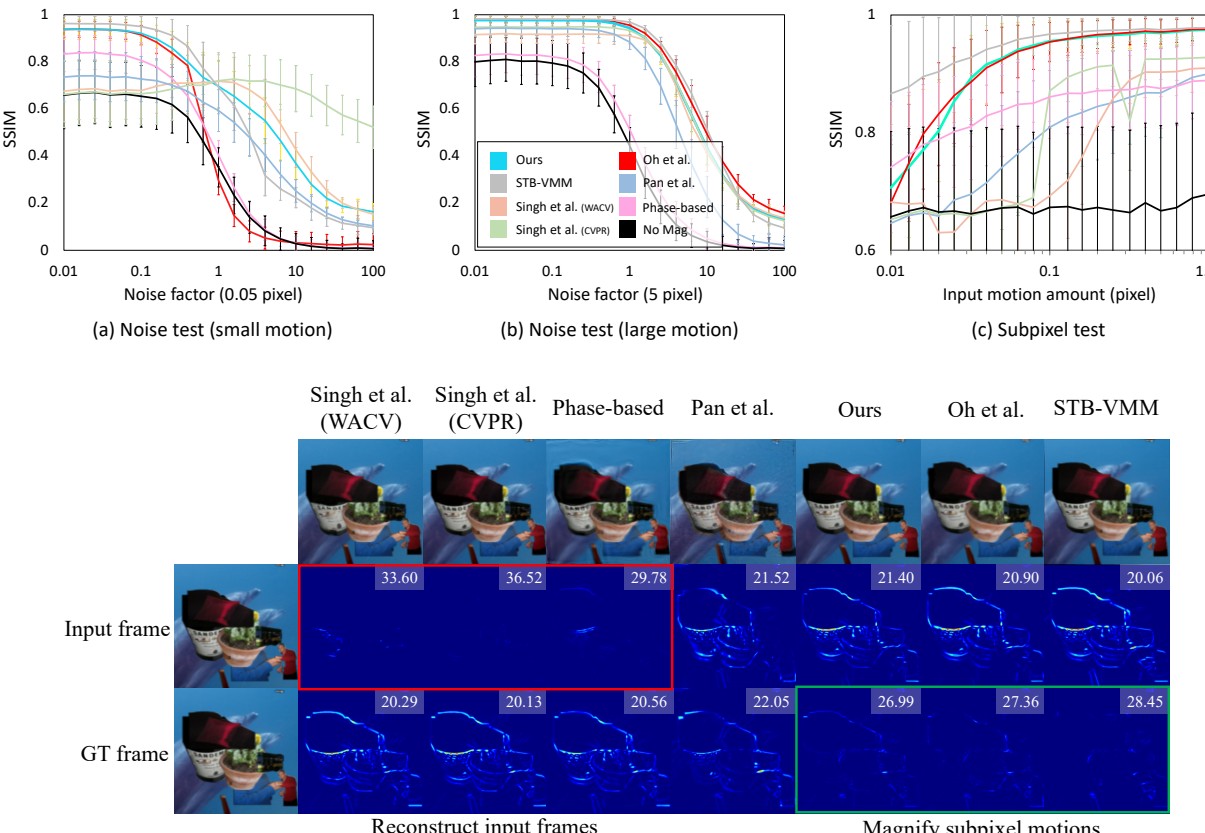

Figure 8: **Noise test and subpixel motion test** [Top] (a, b) Noise test with small motion of 0.05px and large motion of 5px, (c) Sub-pixel motion test. [Bottom] shows an error map and PSNR metric between the ground-truth magnified frame and the magnified frame produced by each method. This example is one of the test pairs for a noise factor of 0.158 in [Top] (a). Our model exhibits strong noise handling, particularly surpassing STB-VMM (Lado-Roigé & Pérez, 2023) in small motion noise tests as the noise factor increases. In contrast, Singh *et al.* (Singh et al., 2023a;b) show weaker performance, often replicating the input image rather than magnifying motion. The subpixel test further confirms the superiority of our model in maintaining magnification quality across all methods.

Pérez, 2023), shows considerably smaller errors in comparison to the GT frames, indicating a more successful magnification of small motions.

The subpixel test further supports this trend. Across most of the evaluated range, our model maintains magnification quality comparable to the baseline (Oh et al., 2018) and STB-VMM (Lado-Roigé & Pérez, 2023), while outperforming other prior methods (Wadhwa et al., 2013; Singh et al., 2023a;b; Pan et al., 2024). At the same time, the lower end of the range, particularly around 0.01 pixel, remains challenging, where STB-VMM appears stronger than both our method and the baseline.

## 6.4 Physical Accuracy

To measure the physical accuracy of various motion magnification methods, we simulate a 21 Hz sinusoidal vibration in a laboratory setting using a vibration generator following the setup of Byung-Ki et al. (2024). An accelerometer attached to the generator measures the peak amplitude of acceleration $(m/s^2)$, which is then converted into sinusoidal displacement waves in both real-world (m) and pixel (px) units using the formula $\mu = a/\omega^2$ where $\omega$ is the vibration frequency. Then, we overlay the converted sinusoidal displacement into the image plane using the formula of pixel displacement $k(t) = \frac{f}{Lv}s(t)$, where $f$ is the focal length, $L$ the distance from the camera to the vibrator, and $v$ the sensor's per-pixel size. The conversion parameters are detailed in Appendix E.

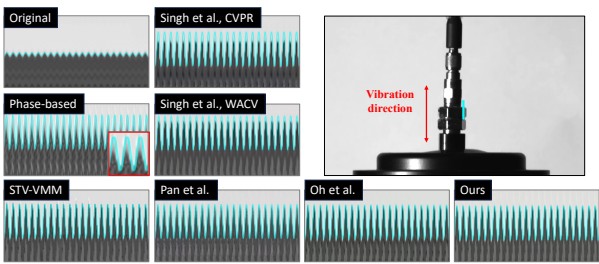

Figure 9: **Physical accuracy on motion magnification.** The vibration simulator is set to generate a periodic vibration with the frequency 21 Hz. Our model captures the frequency of the vibration accurately and magnifies the sinusoidal periodic signal with fewer artifacts, showing comparable quality to the baseline, which suggests physical accuracy of our method. We use Finite Impulse Response (FIR) filter with a frequency band of lower bound 17 Hz and upper bound 25 Hz, and the amplification factor $\alpha$ of 20.

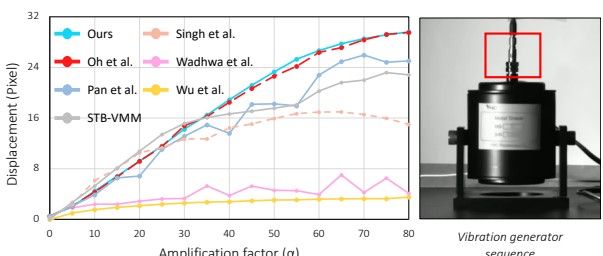

Figure 10: **Comparison of linearity of motion magnification according to amplification factor $\alpha$.** Learning-based methods (Pan et al., 2024; Singh et al., 2023a; Lado-Roigé & Pérez, 2023) including the baseline (Oh et al., 2018) and ours show linear magnification of input motion according to the amplification factor $\alpha$, whereas signal processing based methods including Wu et al. (2012) and Wadhwa et al. (2013) show distorted and attenuated motion magnification. The peak-to-peak displacements are estimated by the masked variant (Ha & Oh, 2023) of Kanade-Lucas-Tomasi tracking algorithm (Tomasi & Kanade, 1991).

As demonstrated in Fig. 9, our method aligns well with the $20\times$ magnified video, preserving the sinusoidal pattern and edge details comparably to the baseline method (Oh et al., 2018). In contrast, the phase-based method (Wadhwa et al., 2013) introduces significant ringing effects and fails to accurately reconstruct the motion. Singh et al. (2023a) also struggle with accurately capturing the sinusoidal signal, indicating inferior separation of motion of interest from the background.

We further verify the linearity between the amplification factor and the resulting motion magnitude. As shown in Fig. 10, signal processing-based methods (Wu et al., 2012; Wadhwa et al., 2013; 2014; Takeda et al., 2020) exhibit substantial attenuation as the amplification factor increases, making faithful physical quantification difficult.[5] In contrast, both our model and the baseline (Oh et al., 2018) demonstrate clear linearity up to an $80\times$ amplification factor, ensuring reliable motion quantification. These results support the potential of our method as a reliable analysis tool.

## 7 Conclusion

In this paper, we propose the first learning-based motion magnification model running in real-time on Full-HD videos, generating high-quality magnification results. Previous learning-based motion magnification methods either lag behind ours in both generation quality and computational speed, or exhibit comparable generation quality but significantly slower computational speed. These challenges stem from a lack of understanding of which parts and characteristics of a motion magnification model play crucial roles in the target task, which has been barely explored before. We experimentally analyze components of the baseline in terms of their importance and functionality on the human-calibrated quality evaluation metric. Integrating insights from the analyses, we develop a real-time motion magnification model, which achieves a computational speed ranging from at least **2.7×** to a maximum of **34.9× faster** than existing learning-based methods while maintaining perceptually sufficient generation quality. We hope that our analyses can serve as a guide for developing learning-based motion magnification model in the future. Moreover, since the reported speed-up is achieved through structural design without specific software-level optimization, further gains

---

[5]Comparison across method families should be interpreted with care. For signal processing-based methods, the same amplification factor does not necessarily produce the same observed displacement due to approximation limits under large motion or amplification (Wu et al., 2012). By contrast, learning-based methods are trained with targets constructed for specified amplification levels (Oh et al., 2018; Singh et al., 2023a;b; Lado-Roigé & Pérez, 2023; Wang et al., 2024a;b).

may be obtained by combining our approach with additional acceleration techniques such as quantization or pruning. Further discussions regarding temporal filtering trade-offs and practical deployment limitations, such as handling large objects and camera motions, are provided in Appendix J.

**Acknowledgments**    This work was supported by Institute of Information & Communications Technology Planning & Evaluation (IITP) grant (No. RS-2025-25443318, Physically-grounded Intelligence: A Dual Competency Approach to Embodied AGI through Constructing and Reasoning in the Real World (30%); No. RS-2024-00457882, National AI Research Lab Project (17.5%); No. RS-2019-II191906, Artificial Intelligence Graduate School Program (POSTECH) (17.5%)), the National Research Foundation of Korea (NRF) grant (No. RS-2024-00358135, Corner Vision: Learning to Look Around the Corner through Multi-modal Signals (17.5%); No. RS-2024-00453301 (17.5%)), and the InnoCORE program of the Ministry of Science and ICT (No. 26-InnoCORE-01), funded by the Korea government (MSIT).

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

## A Quantitative Results on a Semi-realistic Dataset for Learning-based Methods

To address the domain shift from synthetic to real-world data, we provide complementary evidence from two perspectives. First, the vibration-generator experiments in Sec. 6.4 already offer a controlled real-world benchmark on the motion axis, enabling quantitative evaluation of physical accuracy and linearity under real sensor data. Second, since ground-truth (GT) magnified videos are inherently unavailable for real-world scenes, we additionally conduct a proxy evaluation on a Blender-based semi-realistic dataset following Byung-Ki et al. (2024). As shown in Table A1, our model demonstrates the

Table A1: **Magnification quality comparison on a semi-realistic dataset.**

| Method | SSIM ↑ |
|---|---|
| Ours | 0.809 |
| Oh et al. (2018) | 0.809 |
| Singh et al. (2023b) (D1) | 0.758 |
| Singh et al. (2023b) (D2) | 0.749 |
| STB-VMM (Lado-Roigé & Pérez, 2023) | 0.804 |
| EulerMormer (Wang et al., 2024a) | 0.772 |

best magnification quality among the compared methods. We also observe that all learning-based motion magnification methods exhibit a performance drop on the semi-realistic data compared to the synthetic validation set, which is restricted to 2D motion. As discussed in Byung-Ki et al. (2024), this degradation is primarily due to complex 3D motions and more realistic rendering effects. Taken together, the real-data vibration-generator results and the additional semi-realistic evaluation provide complementary evidence on generalization beyond the synthetic training distribution.

## B Temporal Stability

To complement the qualitative temporal coherence observed in the x-t slices of Figure 7 and the supplementary video, we quantitatively evaluate long-term temporal stability by measuring the standard deviation of pixel intensities in stationary regions. Since these regions do not contain actual motion, their temporal variation is primarily due to sensor noise, and a robust motion magnification model should avoid amplifying such variations into vis-

Table A2: **Temporal stability comparison on real-video samples. Bold** and underlined values indicate the best and second-best results, respectively.

| Method | Real Videos | | | Total |
|---|---|---|---|---|
| | AC2 | Column | Baby | |
| Ours | 0.0057 | 0.0033 | **0.0080** | **0.0057** |
| Phase-based (Wadhwa et al., 2013) | 0.0094 | 0.0074 | 0.0096 | 0.0088 |
| Oh et al. (2018) | 0.0075 | 0.0041 | 0.0099 | 0.0072 |
| Singh et al. (2023a) (D1) | 0.0059 | 0.0034 | 0.0088 | 0.0060 |
| Singh et al. (2023b) (D1) | **0.0055** | 0.0035 | 0.0081 | **0.0057** |
| Pan et al. (2024) | 0.0064 | 0.0039 | 0.0086 | 0.0063 |
| STB-VMM (Lado-Roigé & Pérez, 2023) | 0.0060 | 0.0051 | 0.0088 | 0.0066 |
| EulerMormer (Wang et al., 2024a) | 0.0058 | **0.0029** | 0.0085 | 0.0058 |

ible flickering. This metric therefore serves as a direct measure of unwanted artificial flickering. The metric is computed in the normalized [0,1] range, where lower values indicate better stability. As shown in Table A2, our method achieves the lowest average standard deviation, tying with Singh et al. (2023b) (D1) and outperforming or matching other state-of-the-art methods.

## C Generality of Learning-to-remove Framework

### C.1 Applying for image super resolution model

We apply our learning-to-remove method to an image super resolution model to verify its generality. The MDSR (multi-scale super resolution) model (Lim et al., 2017) mainly composes of 1) three scale specific pre-processing branches (*i.e.*, *an encoder*) which have two residual blocks respectively and 2) the main branch (*i.e.*, *a decoder*) which has 16 residual blocks and processes the representation regardless of the scale. As shown in Table A3,

Table A3: **Our framework applied to image super resolution task.** We eliminate the residual blocks of the encoder and decoder module in the multi-scale SR (MDSR) network, respectively.

| Model | PSNR [dB] ↑ | | | Params [M] ↓ |
|---|---|---|---|---|
| | 2× | 3× | 4× | |
| **MDSR** | 35.61 | 31.65 | 29.57 | 3.23 |
| - Encoder | 35.61 | 31.64 (-0.01) | 29.57 | 2.00 (-38.1%) |
| - Decoder | 34.93 (-0.68) | 31.02 (-0.63) | 28.99 (-0.58) | 2.05 (-36.5%) |

Table A4: **Component removal in attention-based blocks for each module of EulerMormer (Wang et al., 2024a).** We experiment removal of attention-based blocks module-by-module. All experiments are conducted using pre-trained weights. ✗ indicates completely removed, and △ denotes a few components survive.

| Module | Block | # Params [M] ↓ | FLOPs [G] ↓ | Quality Metric | |
|---|---|---|---|---|---|
| | | | | SSIM ↑ | LPIPS ↓ |
| Original | – | 1.51 | 56.0 | 0.984 | 0.107 |
| Encoder | ✗ | 1.35 (-0.16) | 50.2 (-5.8) | 0.982 (-0.002) | 0.116 (+0.009) |
| Manipulator | ✗ | 1.43 (-0.08) | 53.1 (-2.9) | 0.982 (-0.002) | 0.119 (+0.012) |
| Decoder | △ | 0.54 (-0.97) | 20.3 (-35.7) | 0.952 (-0.032) | 0.267 (+0.160) |

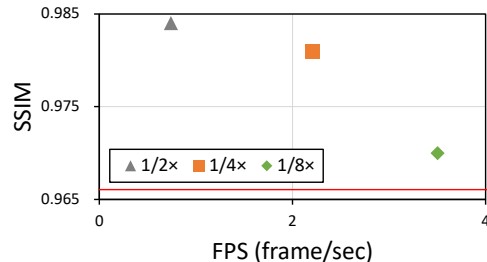

Figure A1: **Quantitative results varying the spatial resolution on Euler-Mormer (Wang et al., 2024a).** $\frac{1}{2}\times$ denotes the original model. We plot the SSIM versus FPS trade-off graph for different downsampling factors. The $\frac{1}{4}\times$ downsampling gains FPS with marginally sacrificing SSIM.

we discover that no noticeable PSNR drop occurs in the model while achieving notable parameter reduction, even if the residual blocks in the encoder of MDSR are totally removed. On the other hand, removing all 16 residual blocks in the decoder leads to significant quality degradation while reducing parameters relatively smaller than removing the encoder module. From these observations, we can investigate the redundancy of neural components of each module in the image super resolution model; *i.e.*, the encoder, whose functionality is scale specific pre-processing, can deal with the task without any heavy residual blocks. These results demonstrate the potential applicability of our learning-to-remove method beyond the motion magnification task.

### C.2 Applying our framework on recent architectures

We further apply the learning-to-remove method to two recent models—STB-VMM (Lado-Roigé & Pérez, 2023) and EulerMormer (Wang et al., 2024a). As you can see in Table A4 and Table A5, we observe the same pattern as in the main paper: the encoder is less sensitive to removal than the decoder. Additionally, for EulerMormer, we conduct spatial bottleneck experiments. From Figure A1, we can find that the $\frac{1}{4}$ resolution setting provides a favorable trade-off, consistent with the observation from Sec. 5.2[6]. These results highlight that our insights extend beyond a single baseline and remain valid across different architectures.

## D Comparison to NAS Results

We conduct experiments to find an efficient model by adopting one of the one-shot NAS methods (Guo et al., 2020), following the same evaluation setting described in Sec. 4.1. The search space used in the NAS experiments includes the kernel size and the number of residual blocks in the encoder and decoder. We set the constraints for each NAS experiment as [max layer depth of the encoder, of the decoder, channel dimension of the decoder, max FLOPs]: (a) [7,9,8,10G], (b) [7,9,16,10G], (c) [7,9,32,20G], and (d) [7,18,8,10G].

The SSIM and FLOPs of the NAS models are compared with our model in Fig. A2. All the architectures found by NAS show significant quality degradation, lying on the poor trade-off between SSIM and FLOPs compared to our proposed model.

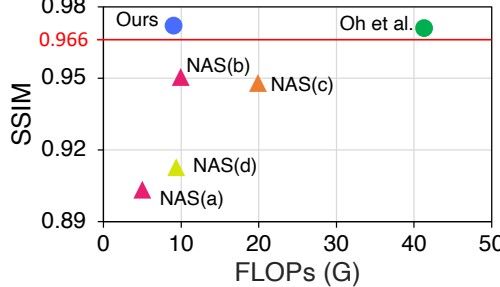

Figure A2: **Quantitative comparison of our proposed model and NAS-searched models.**

---

[6]STB-VMM (Lado-Roigé & Pérez, 2023) is excluded from spatial bottleneck experiment due to memory footprint issues

Table A5: **Component removal in attention-based blocks for each module of STB-VMM (Lado-Roigé & Pérez, 2023).** We experiment removal of attention-based blcoks module-by-module (*i.e.*, the encoder, the manipulator, and the decoder). All the experiments are conducted using a pre-trained weights. The cross mark ✗ indicates that the components are completely removed, and the triangle symbol △ denotes that a few components are survived. In STB-VMM, the computational cost (FLOPs) is heavily concentrated in the decoder's upsampling convolution layer, accounting for 94.8%. This concentration stems from the layer's inefficient design (`ConvTranspose2d` with $8 \times 8$ transposed convolution). To facilitate interpretation of the results, we also report FLOPs excluding the upsampling layer (see the *FLOPs wo. up_conv [G]* column in the table.

| Module | Block | # Params [M] ↓ | FLOPs [G] ↓ | | Quality Metric | |
| --- | --- | --- | --- | --- | --- | --- |
| | | | w. up_conv | wo. up_conv | SSIM ↑ | LPIPS ↓ |
| Original | − | 31.3 | 366.9 | 19.0 | 0.978 | 0.178 |
| Encoder | ✗ | 18.6 (-12.7) | 361.8 (-5.1) | 13.9 (-5.1) | 0.978 | 0.174 (-0.004) |
| Manipulator | ✗ | 30.2 (-1.1) | 364.6 (-2.3) | 16.7 (-2.3) | 0.978 | 0.174 (-0.004) |
| Decoder | △ | 24.9 (-6.4) | 364.4 (-2.5) | 16.5 (-2.5) | 0.962 (-0.016) | 0.247 (+0.069) |

Table A6: **Hyperparameters for processing with temporal filters.** FIR stands for Finite Impulse Response filter.

| Sequence | Amp. Factor | FPS | Temporal Filter | Freq. Band (Hz) |
| --- | --- | --- | --- | --- |
| *crane* | 75× | 24 | FIR | 0.2 - 0.25 |
| *drum* | 10× | 1900 | FIR | 74 - 78 |
| *guitar* | 25× | 600 | FIR | 72 - 92 |
| *Wheel1* | 20× | 120 | FIR | 20 - 30 |
| *baby* | 20× | 30 | Butterworth | 0.01 - 1 |

Table A7: **Hyperparameters for standard mode.**

| Sequence | Amp. Factor | mode |
| --- | --- | --- |
| *baby* | 20 × | Static |
| *AC2* | 8 × | Static |
| *column* | 7 × | Dynamic |
| *gun* | 8 × | Dynamic |

Table A8: **Hyperparameters for acquiring pixel displacement.**

| Hyperparameters | Unit | Value |
| --- | --- | --- |
| Vibration frequency $\omega$ | Hz | 21 |
| Peak amplitude of acceleration $a$ | m/s$^2$ | 4.11 |
| Camera-to-vibrator distance $L$ | m | 2 |
| Focal length $f$ | mm | 100 |
| Per-pixel sensor size $v$ | $\mu$m | 5.86 |

Specifically, NAS (b) shows the best SSIM among all the NAS models but still is below 0.966, the visual acceptability threshold. These results support that an efficient yet effective model could not be found straightforwardly through standard search methods.

Figure A3 shows the searched architectures which achieve the best quality under fixed FLOPs constraints. The search space of the NAS includes the kernel sizes and allows skipping the residual block. When we increase the maximum number of residual blocks (*i.e.*, layer depth) in the search space, a few NAS experiments fail to converge. We find that even the best architecture searched on the trainable configuration cannot approach the optimal trade-off between the quality (SSIM) and the computational costs (FLOPs) in comparison to our proposed architecture.

## E  Hyperparameter Setup

We give the parameters we used for each video. We conduct both experimental cases, motion magnification with temporal filter and without temporal filter to verify the compatibility and generality of our method. We set the amplification factor $\alpha$, temporal filter type and frequency band as given in Table A6 and Table A7. We also set hyperparameters for acquiring pixel displacement in the physical accuracy test as given in Table A8.

## F  Encoder with Deeper Layer

We observe that a deeper decoder achieves better generation quality due to its need for a large receptive field (see Sec. 5.3 of the main paper). To evaluate whether a larger receptive field in the encoder enhances magnification quality, we conduct an additional experiment: training the encoder with increased depth and the decoder with reduced depth. We ensure similar computational costs (FLOPs) across configurations for a fair comparison. As indicated in Table A9, the configurations with a deeper encoder and shallower decoder

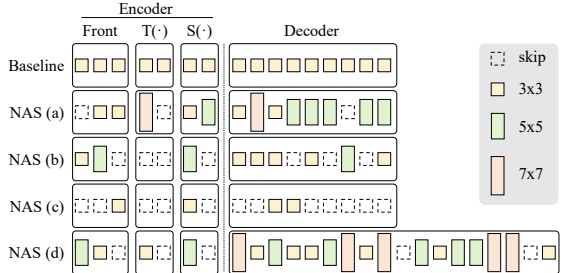

| | Constraints | | | | Results | | |
|---|---|---|---|---|---|---|---|
| **Enc.** | **Dec.** | **Ch.** | **Max. FLOPs [G]** | | **Trainable** | **SSIM** | **FLOPs [G]** |
| | | 8 | 10 | | ✓ | 0.824 | 4.96 |
| | 9 | 16 | 10 | | ✓ | 0.887 | 9.95 |
| | | 32 | 20 | | ✓ | 0.884 | 19.86 |
| 7 | | 8 | 10 | | ✓ | 0.836 | 9.34 |
| | 18 | 16 | - | | ✗ | - | - |
| | | 32 | - | | ✗ | - | - |

Figure A3: **[Left] Found configurations by the NAS experiments, [Right] Constraints and results of the NAS experiments.** NAS (a) - (d) of [Left] correspond to the row 1 - 4 of [Right]. [Left] From the found configurations, we observe that several residual blocks in the decoder are removed, in contrast, our proposed model discards only the residual blocks in the encoder and the manipulator. [Right] Constraints are summarized as follows: Maximum number of residual blocks in the encoder (Enc.) and the decoder (Dec.), fixed channel dimension of the decoder (Ch.) and maximum FLOPs of the architecture (Max. FLOPs). The channel dimension of the encoder is set as half of the channel of the decoder. Training is failed in the two experiments due to gradient exploding. We refer to Guo et al. (2020) for designing the NAS experiments.

Table A9: Effect of increasing encoder layer depth while maintaining FLOPs. The first row corresponds to the baseline (Oh et al., 2018).

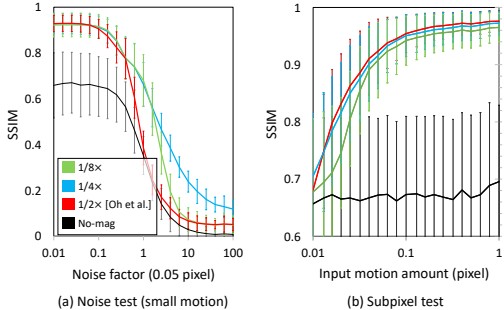

(a) Noise test (small motion)

(b) Subpixel test

Figure A4: **(a) Noise test and (b) Subpixel motion test for downsampled motion representation.**

| **Enc** | **Dec** | **FLOPs [G]↓** | **# Params [K]↓** | **Quality Metric** | |
|---|---|---|---|---|---|
| | | | | **SSIM ↑** | **LPIPS ↓** |
| **7** | **9** | **41.3** | **967.0** | **0.971** | **0.190** |
| 13 | 8 | 41.7 | 1,004 (+37.0) | 0.970 (-0.001) | 0.193 (+0.003) |
| 22 | 6 | 40.8 | 1,023 (+56.0) | 0.968 (-0.003) | 0.208 (+0.018) |
| 28 | 5 | 41.2 | 1,060 (+93.0) | 0.966 (-0.005) | 0.212 (+0.022) |

exhibit significant degradation in SSIM and LPIPS. This indicates that enlarging the encoder's receptive field does not improve the generation of magnified frames.

## G   Effect of Downsampling Motion Representations

Considering that robustness to very small motions and noise can be sensitive to changes in spatial resolution, we evaluate the downsampled representations using two test scenarios, *i.e.*, the noise test and the subpixel test, following the evaluation setup described in Sec. 3.2. As shown in Fig. A4, the $\frac{1}{4}\times$ downsampled representation achieves the best SSIM in the noise test, while showing only marginal attenuation relative to the $\frac{1}{2}\times$ downsampled representation for subpixel motions below 1 pixel. Together with the trade-off analysis in Sec. 5.2 , these results support the $\frac{1}{4}\times$ downsampled representation as a favorable design choice for the settings evaluated in this work.

## H   Calibration on Quality Evaluation Metric

When we evaluate the models using image similarity measures, *e.g.*, SSIM, a remaining challenge is that we are still not sure how well the metrics align with human perception. Due to the absence of real ground truth videos, prior work follows the evaluation method suggested by Oh et al. (2018), where similarity metrics including SSIM are measured with a synthetic evaluation subset. However, due to the small domain gaps between the training and evaluation data, the SSIM values obtained from the synthetic evaluation are biased

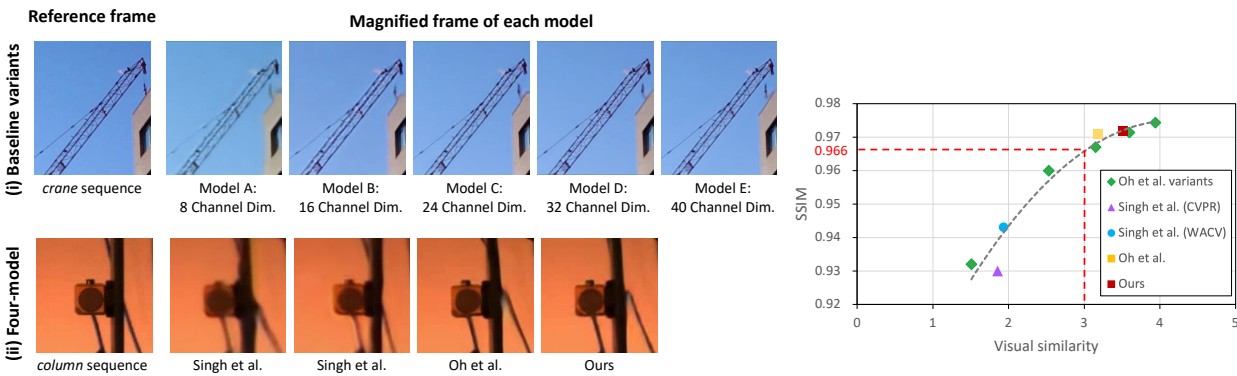

Figure A5: **[Left] Samples of the magnified frame from each model used for the human study. [Right] Relationship between SSIM and visual similarity scored by humans.** We conduct a two-fold user study: (i) variants of the baseline model (Oh et al., 2018), and (ii) a four-model (Singh et al., 2023a;b; Oh et al., 2018) comparison, including ours. We develop four architecture variants (Models A, B, C, and E) by altering the entire channel dimensions of the baseline model (Oh et al., 2018) (Model D) from 8 to 40. These models have different SSIM scores (ranging from 0.9320 to 0.9744) on the synthetic dataset according to channel dimensions. We use them as reference models for calibration to see the relationship between the metric in the synthetic data and the perceptual quality in real data. We ask 20 participants to assess the visual similarity between the input reference frame and the magnified frames generated by each models. We observe that a score of "3" (Adequate Similarity) corresponds to an SSIM of 0.966 by fitting a quadratic trend line to the (SSIM, human scored visual similarity). Based on this, we set a visual threshold of above 0.966 as the primary criterion for determining the acceptability of a proposed architecture, establishing a standard for visually acceptable quality in motion magnification models.

towards being very high. Moreover, models with subtle differences in high SSIM values (*e.g.*, 0.93 and 0.96) on the synthetic evaluation exhibit noticeable quality differences in real videos as perceived by humans. To address this problem, we attempt a simple calibration of the quality evaluation metric, *i.e.*, SSIM, by conducting a human study on the relationship between SSIM measured on synthetic data and visual quality for real data through real video examples.

We conduct a two-fold user study: (i) variants of the baseline model (Oh et al., 2018), and (ii) a comparative study across four models, including ours. We first train four architecture variants (Models A, B, C, and E) by altering the entire channel dimensions of the baseline model (Oh et al., 2018) (Model D). These models show different SSIM scores (ranging from 0.9320 to 0.9744) on the synthetic dataset. Using these models, we magnify real video sequences (*e.g.*, crane and column in Fig. A5 [Left]).

To extend the evaluation beyond baseline variants, we also conduct human study on across several methods. We select models whose SSIM on the synthetic dataset fall in a mid-range—neither too low nor too high (Singh et al., 2023a;b; Oh et al., 2018) including ours proposed model.

Prior to the human studies, participants are trained on examples of the motion magnification task to become acquainted with the task. Then, we ask 20 participants to rate visual similarity between the input reference frame and the magnified frame generated by each case, with a description "Please rate the visual similarity between the input reference image and the magnified frames on a scale of 0 to 5, where each score corresponds to No, Minimal, Moderate, Adequate, Strong, and Perfect similarity to the reference, respectively." Each study involves 10 samples; some samples differ across studies because certain models could not use temporal filtering.

The results from the study indicate that a score of "3" (Adequate Similarity) corresponds to an SSIM of approximately 0.966 (see Fig. A5 [Right]). Based on this observation, we establish a *visual threshold* level of above *0.966* as a primary criterion for determining the acceptability of a proposed architecture. This allows

Table A10: **Maximum input resolution each GPU can process in real time ($\geq$ 24 FPS).** The maximum resolution is selected from the set $\mathcal{R} = \{240\text{p}, 360\text{p}, 480\text{p}, 540\text{p}, 720\text{p}, 1080\text{p}\}$.

| GPU | Max Resolution $\geq$ 24 FPS | |
|---|---|---|
| | Baseline | Ours |
| RTX 3090 (server) | 540p | 1080p |
| RTX 3070 (desktop) | 360p | 720p |
| RTX 2080 Super (laptop) | 240p | 540p |

Table A11: **Throughput at Full-HD (1080p) resolution.** Speed-up is the performance gain of our method over the baseline (Oh et al., 2018).

| GPU | FPS @ 1080p | | Speed-up |
|---|---|---|---|
| | Baseline | Ours | |
| RTX 3090 (server) | 9.4 | 25.7 | 2.73$\times$ |
| RTX 3070 (desktop) | 3.9 | 10.6 | 2.72$\times$ |
| RTX 2080 Super (laptop) | 2.8 | 7.6 | 2.71$\times$ |

Table A12: **Comparison of GPU memory usage (GB) at 1080p resolution.**

| Method | Memory (GB) |
|---|---|
| Ours | 2.34 |
| Oh et al. (2018) | 4.05 |
| Singh et al. (2023b) (D1) | 3.10 |
| Singh et al. (2023b) (D2) | 3.10 |
| STB-VMM (Lado-Roigé & Pérez, 2023) | 3.79 |
| EulerMormer (Wang et al., 2024a) | 13.39 |

Table A13: **Comparison of execution latency (ms) at 1080p resolution.** The reported latency denotes pure GPU execution time, highlighting speedup achieved solely through architectural efficiency.

| Method | Module-wise latency (ms) | | | Total latency (ms) |
|---|---|---|---|---|
| | Encoder | Manipulator | Decoder | |
| Oh et al. (2018) | 16.83 | 7.64 | 79.06 | 103.53 |
| Ours | 1.35 | 3.53 | 34.48 | 39.36 |

us to establish a threshold for visually acceptable generation quality that a motion magnification model should adhere to.

# I Computation speed test on various hardware

As summarized in Table A11, we benchmark our model on three GPUs—RTX 3090 (server), RTX 3070 (desktop), and RTX 2080 Super (laptop), consistently achieving a 2.7$\times$ speed-up and reaching 25.7 FPS at 1080p (on the server). As summarized in Table A10, the model sustains real-time motion magnification ($\geq$24 FPS) at substantially higher input resolutions than Oh et al. (2018) on all devices, increasing the maximum processable pixels by 4$\times$ (RTX 3090/3070) to 5$\times$ (RTX 2080 Super).

We also highlight the practical deployment advantages of our method against heavy transformer-based models. Figure A6 illustrates the processing speed of STB-VMM (Lado-Roigé & Pérez, 2023) and EulerMormer (Wang et al., 2024a) as input resolution is progressively reduced. While our model easily achieves 25.8 FPS at 1080p, transformer-based methods fail to provide real-time performance at practical resolutions. EulerMormer requires extreme downscaling to 144p to achieve comparable FPS, inevitably leading to severe spatial quality degradation that is unacceptable for real-world applications. These results confirm that unlike transformer approaches, our architecture scales effectively to Full-HD without sacrificing visual integrity to meet real-time constraints. This experiment is conducted on a workstation equipped with a single NVIDIA RTX 3090 GPU (24GB).

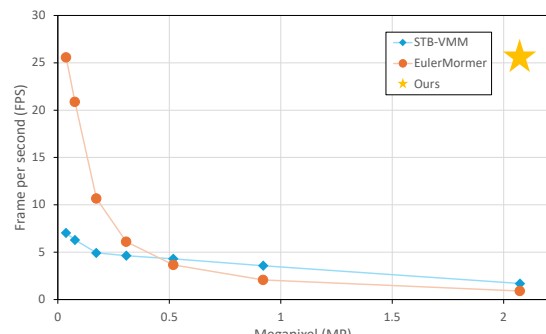

Figure A6: **Processing speed (FPS) under varying input resolutions.**

To verify that our reported acceleration stems from fundamental architectural improvements rather than software-level implementation dependencies, we further analyzed the runtime characteristics of our model. Following experiments are conducted on a workstation equipped with a single NVIDIA RTX 3090 GPU

(24GB). As demonstrated in Table A12, our method requires the lowest average GPU memory (2.34 GB) at 1080p resolution among recent learning-based approaches, highlighting its architectural memory efficiency. Furthermore, we evaluated the pure GPU execution latency per module to understand the source of our speed-up. As detailed in Table A13, the profiling results indicate that the overall acceleration is driven by significant latency drops across all individual modules. Notably, compared to the baseline (Oh et al., 2018), our encoder latency is reduced from 16.83 ms to 1.35 ms, and the decoder latency decreases from 79.06 ms to 34.48 ms. These detailed runtime breakdowns comprehensively validate the architectural efficiency of our proposed method.

Beyond offline benchmarks, we implemented a live webcam demo (540p) on the RTX 2080 Super laptop, which achieves real-time performance in an interactive setting. We include a live demonstration video of this code in the supplementary video. This demonstrates that the efficiency gains translate to practical usage scenarios.

## J   Discussion and Limitations

In this section, we discuss the practical trade-offs, deployment challenges, and limitations of our approach. In particular, we cover temporal filtering, large object and camera motion, complex occlusion, and extreme amplification factors.

**Strengths of structural efficiency.**   We focus on developing a structurally efficient architectural design. Thus, our approach is orthogonal to dynamic pruning or quantization, which could be applied on top of our model if desired. For example, if quantization achieves a $2\times$ acceleration, the overall speed-up would be multiplied, yielding approximately $2.7 \times 2 = 5.4\times$. This highlights that the architectural design we focus on provides a significant contribution.

**Advantages of learned representations.**   In our experiment, the encoder is simplified substantially without a significant loss in performance. This does not mean that learned spatial representations are unnecessary. Rather, compared with classical hand-designed decompositions, the main advantage of a learned representation is that it can learn filters directly optimized for the motion-magnification objective, rather than being fixed by analytical assumptions. This can yield a more compact and task-specific representation than general-purpose classical decompositions. Oh et al. (2018) also report that such learned spatial representations lead to less ringing artifact and better noise characteristics than previous classical approaches.

**Trade-offs of using temporal filter.**   While temporal filtering is beneficial for suppressing noise and unwanted motion so that amplification can focus on the motion-of-interest and yield smoother results, it also increases computational cost and may unintentionally attenuate or remove small parts of the target motion. Therefore, when applying temporal filtering, it is crucial to choose the frequency band and filter type appropriately for the given scenario.

**Lack of in-depth investigation on training pipeline.**   While we provide a comprehensive analysis on the baseline, some configuration choices (e.g., channel reduction and layer depth) may appear architecture-specific. In this context, we would like to position our results as design guidelines: careful treatment of spatial bottlenecks, channel widths, and module sensitivities can systematically improve efficiency across motion-magnification models. There is also a need for a more in-depth examination of training datasets and pipelines. Finally, although our model achieves real-time throughput, further optimization is needed for high-FPS capture scenarios.

**Large object motion.**   For large object motion, magnifying such motion is not the scope of video motion magnification, as the primary goal of this technique is to reveal and visualize subtle motions that are otherwise difficult to perceive. In practice, large motions can be suppressed using complementary techniques such as temporal frequency filtering, depending on the frequency band of interest. More advanced temporal filtering approaches, such as Video Acceleration Magnification (Zhang et al., 2017) and Jerk-Aware Video Acceleration Magnification (Takeda et al., 2018), have also been proposed to specifically address large-magnitude motions.

**Camera motion.** For camera motion, since video motion magnification inherently amplifies all types of motion, camera motion is also magnified. As shown in Figure A7, even when the subjects are static, the camera motion itself is magnified. Unlike large object motion, this cannot be separated from object motion in current 2D motion magnification methods, and we therefore consider it a limitation of the existing framework. Recently, a 3D motion magnification approach (Feng et al., 2023) extended the setting to handle moving cameras using time-varying radiance fields, but it require multi-view data and NeRF-based representations, making the setup fundamentally different from 2D video motion magnification.

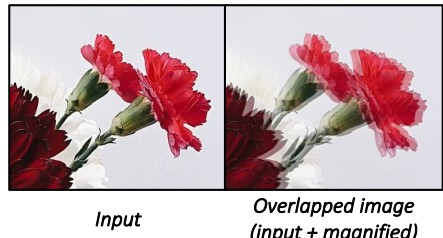

*Input*     *Overlapped image (input + magnified)*

Figure A7: **Camera motion magnified case.**

**Complex occlusion and disocclusion.** Motion magnification models typically operate without explicit depth-level information, making them inherently vulnerable to complex occlusion scenarios. However, unlike traditional signal-processing methods, learning-based models implicitly perform background inpainting near the edges of moving objects. This inherent feature helps the network handle disocclusions to some extent, though highly complex occlusions remain a challenging issue. As demonstrated in Figure A8, our approach is more effective at inpainting disoccluded background regions near edges compared to the Phase-based method (Wadhwa et al., 2013).

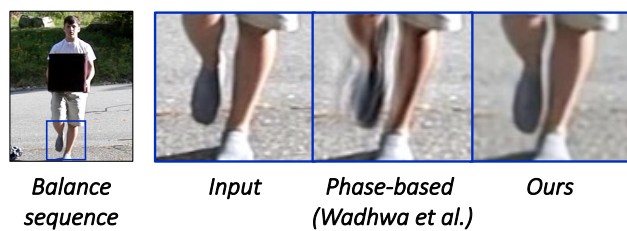

*Balance sequence*     *Input*     *Phase-based (Wadhwa et al.)*     *Ours*

Figure A8: **Comparison on background inpainting near edges.**

**Extreme amplification factors.** As we can see from Fig. 10 in the main manuscript, while the magnification magnitude generally increases linearly, it begins to attenuate at large amplification factors (e.g., exceeding 50). This attenuation is a shared limitation among learning-based methods, as they are bounded by the maximum pixel displacement (e.g., $\leq 30$ pixels) and the maximum amplification factor ($\leq 100$) present in standard synthetic training data (Oh et al., 2018). As shown in Figure A9, severe texture degradation is observed for extreme amplification factors that significantly exceed the training range. Additionally, the finite receptive field of the architecture restricts extreme spatial shifts, and excessively large factors may push magnified regions entirely out of the frame boundaries.



*Input*     *Magnified ($\alpha = 10$)*     *Magnified ($\alpha = 1,000$)*

Figure A9: **Failure case on extreme amplification factor.**

