# OpenReview forum: "Revisiting Learning-based Video Motion Magnification for Real-time Processing"
_TMLR — Decision pending for TMLR_

### Review · Reviewer_zLek · 2026-02-15

**Summary Of Contributions:**

This paper achieves real-time Full-HD video motion magnification. It systematically analyzes Oh et al. (2018)'s model and finds that the encoder can be linearized, a 1/4× spatial bottleneck helps efficiency, and deeper-narrower decoders maintain quality. The model reaches 25.8 FPS at 1080p with comparable quality (SSIM 0.972). The paper proposes a clean analysis framework, a human-calibrated quality threshold, and thorough experiments. However, individual design choices are just standard model compression techniques.

**Audience:**

Yes

**Audience Explanation:**

The paper is relevant to researchers in motion magnification and efficient architectures.

**Broader Impact Concerns:**

No concerns

**Claims And Evidence:**

Yes

**Claims Explanation:**

The main claim is well supported by the progressive ablations. The impulse response visualizations also support the claim.

**Requested Changes:**

- There is still a significant gap between the proposed method and EulerMormer (SSIM/LPIPS: 0.972/0.163 vs. 0.984/0.107). Please discuss whether this gap is significant or not, and reframe the claim "comparable generation quality".
- The perceptual loss in Sec. 5.4 is a training change, not an architectural insight. I wonder whether the ablation of architectural analysis also includes this perceptual loss as part of the architectural changes.

---

> ### Author Response · Authors · 2026-04-15
> **Response to Reviewer zLek’s Review**
>
> ## **Response to Reviewer zLek’s Review**
>
> We thank Reviewer zLek for recognizing our clean analysis framework, the introduction of a human-calibrated quality threshold, and our thorough experiments. Our detailed responses to the raised questions or concerns follow below. All changes are marked in blue within the revised manuscript.
>
> ---
> ### **Q1: Reframing the claim on generation quality compared to EulerMormer**
>
> We tone down the phrase “comparable generation quality,”  by following the reviewer’s suggestion. We have updated the text in the abstract, Sections 1 and 7, from “comparable generation quality” to “perceptually sufficient generation quality.” This change is justified as we established a human-calibrated visual threshold (SSIM of 0.966), which our method satisfies (SSIM of 0.972).
>
> ---
> ### **Q2: Clarification on perceptual loss in architectural analysis**
>
> The perceptual loss in Section 5.4 is indeed a training modification rather than an architectural component. We clarify that all architectural analyses in Sections 5.1-5.3 were conducted without the perceptual loss, ensuring that the reported results purely reflect the effect of architectural changes. The perceptual loss is introduced only in Section 5.4, where we present the final model. To avoid any confusion, we have updated Figure 5 in Section 5.4 to explicitly distinguish this training change from the architectural modifications.

---

### Review · Reviewer_1dVp · 2026-02-15

**Summary Of Contributions:**

The paper systematically revisits the first learning-based video motion magnification model and identifies redundant architectural components through a learning-to-remove framework. Based on controlled architectural analyses (spatial bottlenecks and depth–channel trade-offs), it proposes a lightweight CNN design that significantly accelerates inference. The resulting model achieves real-time Full-HD performance (25.8 FPS) while maintaining comparable magnification quality and physical linearity to prior state-of-the-art methods

**Audience:**

Yes

**Audience Explanation:**

The paper addresses a practical bottleneck in learning-based video processing—real-time inference at high resolution—which is relevant to researchers working on efficient deep architectures, video enhancement, and model compression. Its structured component-removal methodology may also interest researchers studying neural architecture simplification and redundancy analysis. Additionally, the empirical findings on depth–width trade-offs under constant FLOPs provide insights that extend beyond motion magnification.

**Broader Impact Concerns:**

This work mainly focuses on improving efficiency for video motion magnification and does not introduce fundamentally new capabilities that would substantially increase misuse risks. The potential applications (e.g., structural monitoring or medical signal visualization) are generally beneficial and practical. That said, as with any video manipulation technique, there is some inherent risk of misuse for generating misleading visual content, although this does not appear to be significantly exacerbated by the contributions of this paper.

**Claims And Evidence:**

Yes

**Claims Explanation:**

The core contribution is not a new motion formulation but a structured architectural re-design grounded in empirical analysis. The authors provide ablation-driven evidence to justify each structural modification, rather than performing heuristic pruning. The final outcome demonstrates a meaningful deployment-level improvement: the first learning-based model achieving real-time performance on Full-HD inputs without severe quality degradation

**Requested Changes:**

1. The model is trained entirely on the synthetic dataset introduced by Oh et al. (2018) , and while qualitative examples on real videos are presented, there is no quantitative evaluation demonstrating robustness under domain shift. Given that motion magnification is often applied in real-world infrastructure monitoring, biomedical analysis, or mechanical diagnostics, it would significantly strengthen the paper to include quantitative validation on real-world data or controlled physical benchmarks. At minimum, a more rigorous discussion of expected generalization limitations would be valuable.

2. Although the model processes consecutive frames and is evaluated primarily using per-frame SSIM and LPIPS, there is no systematic assessment of long-term temporal stability or flickering artifacts across extended sequences. Since the paper emphasizes real-time deployment, temporal coherence becomes particularly important in practical use. Including temporal consistency metrics, optical-flow-based stability analysis, or a user study focused on temporal smoothness would strengthen the evaluation.

3. The reported speed improvements are substantial, reaching up to 34.9× faster than prior learning-based methods. However, the analysis primarily attributes this to architectural simplification and FLOPs reduction. A more detailed breakdown of runtime characteristics (e.g., per-module latency, memory bandwidth considerations, and GPU utilization profiling) would clarify whether the acceleration gains are fundamentally architectural or partially implementation-dependent.

4. While the paper includes noise and subpixel motion tests, it does not clearly analyze limitations under challenging conditions such as large camera motion, complex occlusion/disocclusion, or extreme amplification factors. A dedicated subsection outlining typical failure modes, along with visual examples and explanations, would provide a more balanced and practically informative evaluation.

5. The comparison with transformer-based models demonstrates a strong real-time advantage for the proposed CNN architecture, but the current evaluation does not normalize for latency or FLOPs across methods. Providing comparisons under equal computational budgets or discussing how transformer-based models degrade under real-time constraints would make the speed–quality trade-off analysis more convincing and comprehensive.

---

> ### Author Response · Authors · 2026-04-15
> **Response to Reviewer 1dVp’s Review (for Q1)**
>
> ## **Response to Reviewer 1dVp’s Review**
>
> We thank Reviewer 1dVp for recognizing our structured component-removal methodology, the empirical insights on depth-width trade-offs, and the practical value of our real-time Full-HD performance. Our detailed responses to the raised questions or concerns follow below. All changes are **marked in blue** within the revised manuscript. Please note that **all section and figure references in this response correspond to the updated numbering** in the revised version.
>
> ---
> ### **Q1: Quantitative validation on real-world data and expected generalization limitations**
>
> We have added the following experimental results and discussion on generalization limitations to Appendix A of the revised manuscript.
>
> As discussed in Oh et al. (2018), obtaining ground-truth (GT) magnified videos from real data is inherently infeasible, since original videos and their motion-magnified counterparts do not co-exist in real-world scenarios by definition. Nonetheless, to address the reviewer’s concern as rigorously as possible under this constraint, we consider two complementary aspects of the synthetic-to-real gap: **motion** and **texture**.
>
> **Experiment 1 (real data in motion).**
>
> We note that our vibration-generator-based experiments (Figures 9 and 10 in Section 6.4) already provide a controlled real-world benchmark for motion, where the motion frequency can be precisely controlled and real sensor data can be acquired. This setup enables quantitative evaluation of physical accuracy and linearity with respect to the amplification factor under real-world conditions, directly supporting motion-related generalization on real data.
>
> **Experiment 2 (semi-realistic data in texture).**
>
> While Experiment 1 allows us to assess generalization on the motion axis using real data, it does not isolate the texture-related domain shift from synthetic to real scenes. To complement this aspect, we additionally conduct a proxy evaluation on semi-realistic data. Since GT magnified videos are unavailable for real-world scenes, we follow the setup of Byung-Ki et al. (2024) and evaluate on a Blender-based semi-realistic dataset.
>
> | Method | SSIM $\uparrow$ |
> |---|---:|
> | Ours | **0.809** |
> | Oh et al. (2018) | **0.809** |
> | Singh et al. (2023b) (D1) | 0.758 |
> | Singh et al. (2023b) (D2) | 0.749 |
> | STB-VMM (Lado-Roigé & Pérez, 2023) | 0.804 |
> | EulerMormer (Wang et al., 2024a) | 0.772 |
>
> On this benchmark, our model demonstrates the best magnification quality among the compared methods. At the same time, all motion magnification methods exhibit a performance drop relative to the previous evaluation on the synthetic validation set, which is restricted to 2D motion. As discussed in Byung-Ki et al. (2024), this degradation is primarily attributed to complex 3D motions and more realistic rendering effects.
>
> Taken together, these two experiments provide complementary evidence on generalization beyond the synthetic training distribution: the vibration-generator results support real-world generalization on the **motion** axis, while the additional semi-realistic evaluation supports robustness to **texture-related** domain shift and more realistic scene complexity.

---

> ### Author Response · Authors · 2026-04-15
> **Response to Reviewer 1dVp’s Review (for Q2-Q3)**
>
> ### **Q2: Assessment of long-term temporal stability**
>
> We thank the reviewer for this valuable suggestion. We have included a quantitative evaluation of temporal stability in Appendix B of the revised manuscript.
>
> Qualitative observations from the x-t slices in Figure 7 show that our method produces temporally coherent motion magnification with well-preserved and continuous motion trajectories, without noticeable flickering artifacts across frames. This behavior remains stable over extended sequences, as further illustrated in the supplementary video.
>
> To assess flickering artifacts in long-term sequences, we measure the temporal standard deviation of pixel intensities in stationary regions of real videos. Since these regions do not contain actual motion, their temporal variation is primarily due to sensor noise, and a robust motion magnification model should avoid amplifying such variations. This metric therefore serves as a direct measure of undesired temporal fluctuations (i.e., artificial flickering). Under this evaluation (computed in the normalized [0, 1] range, where lower is better), our method achieves the lowest average standard deviation among all compared methods.
>
> | Method | AC2 | Column | Baby | Total |
> |---|---:|---:|---:|---:|
> | Ours | 0.0057 | 0.0033 | **0.0080** | **0.0057** |
> | Phase-based (Wadhwa et al., 2013) | 0.0094 | 0.0074 | 0.0096 | 0.0088 |
> | Oh et al. (2018) | 0.0075 | 0.0041 | 0.0099 | 0.0072 |
> | Singh et al. (2023a) (D1) | 0.0059 | 0.0034 | 0.0088 | 0.0060 |
> | Singh et al. (2023b) (D1) | **0.0055** | 0.0035 | 0.0081 | **0.0057** |
> | Pan et al. (2024) | 0.0064 | 0.0039 | 0.0086 | 0.0063 |
> | STB-VMM (Lado-Roigé & Pérez, 2023) | 0.0060 | 0.0051 | 0.0088 | 0.0066 |
> | EulerMormer (Wang et al., 2024a) | 0.0058 | **0.0029** | 0.0085 | 0.0058 |
>
> ---
> ### **Q3: Detailed breakdown of runtime characteristics**
>
> In response, we have included the following detailed runtime profiling results in Appendix I in the revised manuscript to comprehensively validate our architectural efficiency.
>
>
> To clarify that the acceleration is fundamentally architectural, we conduct additional profiling as suggested by the reviewer. The following experiments are conducted on a workstation equipped with a single NVIDIA RTX 3090 GPU (24GB) and dual AMD EPYC 7452 processors (64 physical cores).
>
>
> **1) GPU memory usage**: At 1080p resolution, our method requires the lowest average GPU memory usage (2.34 GB) among recent learning-based methods, demonstrating high architectural memory efficiency.
>
> | Method | Memory (GB) |
> |---|---:|
> | Ours | 2.34 |
> | Oh et al. (2018) | 4.05 |
> | Singh et al. (2023b) (D1) | 3.10 |
> | Singh et al. (2023b) (D2) | 3.10 |
> | STB-VMM (Lado-Roigé & Pérez, 2023) | 3.79 |
> | EulerMormer (Wang et al., 2024a) | 13.39 |
>
>
>
> **2) Per-module execution latency**: We evaluate the pure GPU execution latency per frame for each module at Full-HD (1080p) resolution, averaged over 360 frames. The profiling results clearly indicate that the overall speed-up is driven by latency drops in each individual module. For instance, compared to the baseline, our encoder latency drops from 16.83 ms to 1.35 ms, and the decoder from 79.06 ms to 34.48 ms.
>
>
> | Method | Encoder (ms) | Manipulator (ms) | Decoder (ms) | Total latency (ms) |
> |---|---:|---:|---:|---:|
> | Oh et al. (2018) | 16.83 | 7.64 | 79.06 | 103.53 |
> | Ours | 1.35 | 3.53 | 34.48 | 39.36 |
>
>
> Finally, note that we did not apply any specific software-level implementation techniques to artificially gain computational speed. As already demonstrated in Table A11 of the manuscript, our model consistently yields a ~2.7$\times$ speed-up over the baseline (Oh et al.) when evaluated on a single GPU across various hardware setups (RTX 3090, RTX 3070, and RTX 2080 Super). This confirms that the gains are robust and not limited to a specific GPU model.
>
> In other words, there remains potential for further improvement through additional optimization techniques. We have explicitly included this point as a discussion in the conclusion section of the revised manuscript.

---

> ### Author Response · Authors · 2026-04-15
> **Response to Reviewer 1dVp’s Review (for Q4-Q5)**
>
> ### **Q4: Limitations under challenging conditions**
>
>
> Building upon our in-depth limitation analysis in Appendix J, we have expanded our discussion in the revised manuscript to address these challenging conditions (e.g., complex occlusion, extreme amplification). This expanded discussion is supported by additional visual samples in the revised manuscript (Figure A7-A9) for a more balanced evaluation.
>
> - **Large camera motion**: As originally discussed in Appendix J, current 2D motion magnification methods inherently amplify all spatial shifts. Therefore, global camera motion is also magnified and cannot be easily decoupled from the target object's motion.
> - **Complex occlusion/disocclusion**: Motion magnification models operate without explicit depth-level information, making them inherently vulnerable to complex occlusion scenarios. However, unlike traditional signal-processing methods, learning-based models implicitly perform background inpainting near the edges of moving objects, which helps them handle disocclusions to some extent (please see Figure A8 in the revised manuscript for visual samples).
> - **Extreme amplification factors**: As shown in our linearity experiment (Figure 10), the magnification magnitude increases linearly but begins to attenuate at factors exceeding 50. This is a shared limitation among learning-based methods, as they are bounded by the maximum pixel displacement (<30 pixels) and the maximum amplification factor (<100) present in the synthetic training data (Oh et al., 2018). Additionally, the architecture’s finite receptive field restricts extreme spatial shifts, and excessively large factors may push magnified regions entirely out of the frame boundaries.
>
> In addition to this revision, we would like to respectfully remind the focus of our work, where we revisit the development of the prior art, completely evaluate our method on the regimes suggested by the prior arts, and clearly demonstrate the superior improvement over them. Tackling the application-level challenges the prior arts had is not the focus of our work. The current manuscript provides a clear discussion of the shared limitation, and we believe the reviewer’s comment is appropriately addressed.
>
> ---
> ### **Q5: Comparison with transformer-based models under equal computational budgets**
>
> We thank the reviewer for these careful suggestions. We have added this speed-resolution trade-off analysis to Appendix I in the revised manuscript to make the comparison more comprehensive.
>
> Ensuring a fair comparison under strictly matched computational budgets across fundamentally different architectures (e.g., CNN-based vs. transformer-based models) requires careful re-tuning of architecture and hyperparameters for each method, which constitutes a non-trivial combinatorial problem. In addition, these models exhibit different scaling behaviors in terms of FLOPs, memory access patterns, and latency, making direct budget matching difficult to interpret in practice.
>
> To provide a practically meaningful comparison, we instead evaluate the speed--quality trade-off under a fixed real-time constraint by progressively reducing the input resolution from 1080p (Full-HD) to 144p. This experiment is conducted on a workstation equipped with a single NVIDIA RTX 3090 GPU (24GB) and dual AMD EPYC 7452 processors (64 physical cores).
> | Model | Required Resolution for ~25 FPS | Max FPS (at 144p) | 1080p FPS |
> |---|---:|---:|---:|
> | **Ours** | **1080p (25.76)** | 153.92 | **25.76** |
> | EulerMormer | 144p (25.58) | 25.58 | 0.90 |
> | STB-VMM | Unreachable | 7.03 | 1.69 |
>
> As shown in the table, STB-VMM cannot reach real-time speed (e.g., > 25 FPS) even at an extremely low resolution of 144p. EulerMormer reaches 25.58 FPS, but it requires a drastic downscaling to 144p, which severely degrades the visual quality so that it can be practically unusable.
> These results indicate that, under realistic deployment constraints, current transformer-based models require a substantial sacrifice in spatial resolution to meet real-time requirements. In contrast, our lightweight architecture achieves real-time performance while maintaining high-resolution (Full-HD) outputs, demonstrating a clear practical advantage.

---

### Review · Reviewer_srT5 · 2026-04-01

**Summary Of Contributions:**

The authors propose modifications to a learning-based video motion magnification method [Oh et al., 2019] to make it process frames more quickly. Specifically, they remove certain network components, increase the spatial downsampling factor, decrease the number of channels while adding layers, and add a perceptual loss to the training objective. They demonstrate qualitative and quantitative improvements (in terms of reconstruction quality and/or computational speed) over previous learning-based and classical methods.

Strengths:
* The result of this work is the learning-based approach to motion magnification that is ready for real-time processing.
* The analysis of redundant components is enlightening, especially the finding that the encoder can be made fully linear.
* The method is well-validated, with ablation studies and comparisons to reasonable baselines.

Weaknesses:
* The contribution is more engineering-based than technically innovative.
* The presentation quality could be improved. The writing is a bit clunky, with grammatical errors throughout. Some figures need visual clarification. For example, Figure 1 needs a legend, and Figure 4 needs labels for the columns of the quantitative results and the red line in the quantitative results.
* I would like the authors to clarify something: Figure 10 shows that different methods will produce different actual magnification factors for the same $\alpha$. In that case, how do you ensure fair comparison across methods?

Minor comments/questions:
* I would have liked to see more analysis of why certain network components can be removed. I wonder why use a learned spatial representation at all rather than a classical spatial representation. What new information has the encoder learned beyond a classical approach like complex steerable filters?
* Is it possible that a 4x reduction in spatial resolution is not appropriate for other types of images? To what extent is this modification only useful for the types of images analyzed in the paper?
* How much extra training time does including the perceptual loss incur?
* Please define acronyms like “FIR.”
* Is it correct to say there’s still a fundamental limit of 0.01 pixel?

**Audience:**

Yes

**Audience Explanation:**

There’s enough interest in the motion magnification topic that an approach for real-time magnification should be useful to the community.

**Claims And Evidence:**

Yes

**Claims Explanation:**

The authors provide adequate ablation studies and baseline comparisons.

**Requested Changes:**

I don’t have any required changes to suggest, but I recommend the authors to address the points that I made under “weaknesses” and “minor comments/questions” to improve the paper.

---

> ### Author Response · Authors · 2026-04-15
> **Response to Reviewer srT5’s Review (for Q1-Q3)**
>
> ## **Response to Reviewer srT5’s Review**
> We thank Reviewer srT5 for recognizing the practical importance of real-time processing, the value of our component analysis, and the overall quality of our ablations and comparisons. Our detailed responses to the raised questions or concerns follow below. All changes are **marked in blue** within the revised manuscript. Please note that **all section and figure references in this response correspond to the updated numbering** in the revised version.
>
> ---
> ### **Q1: The contribution is more engineering-based than technically innovative.**
>
> We respectfully believe that the main contribution of this work is not purely engineering-based. The key contribution is a systematic analysis of how learning-based motion magnification can be made practical for real-time deployment while preserving favorable synthesis quality. Rather than introducing a fundamentally new architecture, we identify which components are essential, which can be simplified, and how efficiency–quality trade-offs can be managed in a principled way. In this sense, our work provides experimentally grounded design guidance as findings, rather than merely model compression.
>
> We believe this is meaningful for the field because the prior learning-based methods demonstrated plausible visual quality, but remained below real-time throughput for Full-HD videos, limiting their use in online applications. Concretely, our analyses show that the encoder can be simplified much more aggressively than the decoder, that a moderate spatial bottleneck offers a favorable efficiency--quality trade-off, and that channel reduction should be balanced with sufficient depth rather than applied naively. These design patterns are not specific to a single baseline: Appendix C.2 shows that similar patterns also hold for recent architectures such as STB-VMM and EulerMormer.
>
> We hope the reviewer will value these aspects and our achievement as contributions.
>
> ---
> ### **Q2: The presentation quality could be improved.**
>
> We apologize for the lack of clarity in the presentation and appreciate the reviewer’s careful suggestions. We have proofread and polished the whole manuscript accordingly. Specifically, we have improved grammar and sentence flow throughout the paper, added a clear legend to Figure 1, added labels in Figure 4, and clarified the red horizontal line in the quantitative trade-off plots as the human-calibrated visual threshold (SSIM = 0.966) introduced in Section 4.2. We believe these changes improve readability and overall presentation of the manuscript.
>
> ---
> ### **Q3: Figure 10 shows that different methods will produce different actual magnification factors for the same alpha. In that case, how do you ensure fair comparison across methods?**
>
> Thank you for this very important question. We have added the interpretation of alpha and the scope of fairness in the comparison in Section 6.4.
>
> We first clarify that alpha is the standard amplification factor, where the target motion is formulated as $(1 + \alpha) \times \delta$ (Wu et al., 2012; Oh et al., 2018). In this sense, alpha is not a relative score, but the intended absolute amplification level. Accordingly, comparison among learning-based methods is fair under the same alpha, because these methods are trained with targets constructed for specified amplification levels and share the same protocol (Oh et al., 2018; Singh et al., 2023a;b; Lado-Roigé & Pérez, 2023; Gao et al., 2022; Wang et al., 2024a;b). Signal-processing-based methods should be interpreted more carefully, since the same alpha may not necessarily yield the same observed displacement due to approximation limits under large motion or large magnification (Wu et al., 2012).
>
> Even for learning-based methods, accurately realizing large amplification remains challenging, especially at high values of the user-specified magnification factor $\alpha$. In practice, there exists a trade-off between induced displacement and magnification quality. As a result, some methods may produce visually high-quality outputs partly because they implicitly generate weaker magnification (i.e., smaller actual displacements) than those user-specified by $\alpha$. This discrepancy can lead to misleading evaluations, where apparent improvements in visual quality are in fact due to under-amplification rather than a true ability to handle large magnification. From this perspective, one of the purposes of Figure 10 is to show that our qualitative comparison is stricter for our model, since it maintains stronger magnification while still showing favorable visual quality.

---

> ### Author Response · Authors · 2026-04-15
> **Response to Reviewer srT5’s Review (for Q4-Q7)**
>
> ### **Q4: Why can certain network components be removed? Why use a learned spatial representation at all, rather than a classical one such as complex steerable filters? What new information has the encoder learned beyond classical approaches?**
>
> We have updated the following discussion in Appendix J.
>
> Certain network components can be removed when simplifying them does not meaningfully degrade the overall motion magnification quality. In our learning-to-remove analysis, this is tested by allowing the network to favor simpler paths while still being optimized for the original task objective. The resulting behavior suggests that the encoder in this architecture can be simplified substantially without a significant loss in performance.
>
> This does not mean that learned spatial representations are unnecessary. Rather, compared with classical hand-designed decompositions, the main advantage of a learned representation is that it can learn filters directly optimized for the dedicated motion-magnification objective, rather than being fixed by analytical assumptions. Specifically, complex steerable filters (CSF) [Freeman and Adelson, 1991] are overcomplete and general-purpose; thus, it has much larger coefficients than alternative orthogonal or necessary counterparts [Freeman and Adelson, 1991]. Also, Oh et al. (2018) already reported that the learned spatial alternative leads to less ringing artifact and better noise characteristics than classical approaches, including CSF. In this sense, our work demonstrates the existence of a more compact and effective learned spatial representation in a linear form. We believe that this is an important finding to report to the research community.
>
> *[Freeman and Adelson, 1991] The Design and Use of Steerable Filters, IEEE Trans. on PAMI 1991.*
>
> ---
> ### **Q5: Is it possible that a 4x reduction in spatial resolution is not appropriate for other types of images? To what extent is this modification only useful for the types of images analyzed in the paper?**
>
> We have updated the following discussion in Section 5.2 and Appendix G.
>
> Although it is difficult to determine whether a 4$\times$ spatial reduction is appropriate for all possible image types, we evaluated this choice from multiple perspectives. In Section 5.2, we showed that the 4$\times$ spatial reduction setting provides a favorable trade-off between quality and computational speed. Although the quality metrics are measured on synthetic data, the threshold used to judge whether the quality drop is acceptable was calibrated through a human study on real videos (Section 4 and Appendix H).
>
> Considering that robustness to very small motions and noise can be sensitive to changes in spatial resolution, we evaluated the downsampled representations using sub-pixel motion and noise tests (Appendix G). As shown in Figure A4, the 4$\times$ spatial reduction setting gives the best result in the noise test, while showing only marginal attenuation relative to the 2$\times$ reduction setting for subpixel motions below 1 pixel. Therefore, rather than claiming universal optimality, we position 4$\times$ spatial reduction as a practical design choice that preserves the key robustness properties while substantially improving efficiency in our evaluation setting.
>
> ---
> ### **Q6: How much extra training time does including the perceptual loss incur?**
>
> We have included the discussion on extra training time in Section 5.4. On a single NVIDIA RTX 3090 GPU, training the proposed model takes 7.2 hours without the perceptual loss and 25.6 hours with the perceptual loss, corresponding to an additional 18.4 hours of training time. Despite this overhead, the training cost remains practically manageable, since the full model can still be trained on a single RTX 3090 GPU within about one day. As also clarified in Section 5.4, the perceptual loss is an optional addition for improving perceptual quality rather than a core contribution of our method.
>
> ---
> ### **Q7: Please define acronyms like “FIR.”**
>
> We thank the reviewer for pointing this out. As indicated in the caption of Figure 9, FIR stands for Finite Impulse Response. To improve clarity, we have also added its definition at its first mention in Section 6.2 and in Table A6.

---

> ### Author Response · Authors · 2026-04-15
> **Response to Reviewer srT5’s Review (for Q8)**
>
> ### **Q8: Is it correct to say there is still a fundamental limit of 0.01 pixel?**
>
>
> We address the issue of a “fundamental limit” from two perspectives: the evaluation range itself and the interpretation of the experimental results. We have clarified these points in Section 4.3 and Section 6.3 in the revised manuscript, respectively.
>
>
> As described in Section 4.3, our sub-pixel evaluation range follows Oh et al. (2018), which likewise uses 0.01 pixel as the smallest tested motion magnitude. We do not interpret this range as a fundamental limit. Rather, we regard it as a practically challenging lower bound in commonly used synthetic evaluation protocols.
>
>
> At the same time, we agree that experimental results should be interpreted carefully. As shown in Fig. 8 [Top] (c), the 0.01-pixel regime remains challenging, and STB-VMM appears stronger than our method and the baseline of Oh et al. (2018) at the lower end of the range. We have revised Section 6.3 to make this point clearer and to provide a more balanced interpretation.

---

### Decision · Action_Editor_X4TU · 2026-06-16

**Recommendation:** Accept as is

**Audience:**

Yes

**Audience Explanation:**

All reviewers indicate the work is a valuable contribution with practical value in real-time video motion magnification, and represents an important milestone albeit in a niche area.  The reviewers also note (and the AE agrees) that the empirical findings on depth–width tradeoffs under constant FLOPs provide insights that extend beyond motion magnification.

**Claims And Evidence:**

Yes

**Claims Explanation:**

All reviewers agree the work is technically sound, backed by sufficiently comprehensive empirical evidence. The revision process was notably productive, leading to more robust evaluations and better understanding of the work, and resolving the main concerns of the reviewers.